# IL-7Rα glutamylation and activation of transcription factor Sall3 promote group 3 ILC development

Benyu Liu[1,2], Buqing Ye[1], Xiaoxiao Zhu[3], Guanling Huang[1,2], Liuliu Yang[1,2], Pingping Zhu[1], Ying Du[1], Jiayi Wu[1,2], Shu Meng[3], Yong Tian [2,3] & Zusen Fan[1,2]

Group 3 innate lymphoid cells (ILC3) promote lymphoid organogenesis and potentiate immune responses against bacterial infection. However, how ILC3 cells are developed and maintained is still unclear. Here, we show that carboxypeptidase CCP2 is highly expressed in common helper-like innate lymphoid progenitors, the progenitor of innate lymphoid cells, and CCP2 deficiency increases ILC3 numbers. Interleukin-7 receptor subunit alpha (IL-7Rα) is identified as a substrate of CCP2 for deglutamylation, and IL-7Rα polyglutamylation is catalyzed by polyglutamylases TTLL4 and TTLL13 in common helper-like innate lymphoid progenitors. IL-7Rα polyglutamylation triggers STAT5 activation to initiate transcription factor *Sall3* expression in common helper-like innate lymphoid progenitors, which drives ILC3 cell differentiation. Moreover, *Ttll4*[−/−] or *Ttll13*[−/−] mice have reduced IL-7Rα polyglutamylation and Sall3 expression in common helper-like innate lymphoid progenitors. Importantly, mice with IL-7Rα E446A mutation have reduced Sall3 expression and ILC3 population. Thus, polyglutamylation and deglutamylation of IL-7Rα tightly controls the development and effector functions of ILC3s.

[1] Key Laboratory of Infection and Immunity of CAS, CAS Center for Excellence in Biomacromolecules, Institute of Biophysics, Chinese Academy of Sciences, Beijing 100101, China. [2] University of Chinese Academy of Sciences, Beijing 100049, China. [3] Key Laboratory of RNA Biology of CAS, Institute of Biophysics, Chinese Academy of Sciences, Beijing 100101, China. Benyu Liu, Buqing Ye, and Xiaoxiao Zhu contributed equally to this work. Correspondence and requests for materials should be addressed to Y.T. (email: ytian@ibp.ac.cn) or to Z.F. (email: fanz@moon.ibp.ac.cn)

Innate lymphoid cells (ILCs) are part of the innate immune system[1, 2]. ILCs reside in the mucosal tissues and respond rapidly to pathogen infection or tissue damage via germ line-encoded receptors[3–5]. ILCs can be categorized into three groups based on their signature effector cytokines analogous to the classification of CD4[+] helper T-cell subsets[6]. Group 1 innate lymphoid cells (ILC1), including natural killer (NK) cells and ILC1s, function in the immune response to intracellular pathogens via secreting interferon-γ (IFN-γ)[7]. Group 2 innate lymphoid cells (ILC2), including natural helper cells, nuocytes and innate helper 2 cells, enhance the resistance to helminth infection through secreting type 2 T helper (Th2) cell cytokines[8, 9]. Group 3 innate lymphoid cells (ILC3), including lymphoid tissue inducer (LTi) cells, natural cytotoxicity receptor positive (NCR[+]) and NCR[−] ILC3s, promote lymphoid organogenesis and potentiate immune responses against bacterial infection, respectively, via producing cytokines IL-17 and IL-22[10–12].

All ILC cells are derived from common lymphoid progenitors (CLPs), which also differentiate to T and B cells[13]. ILC3s, together with other ILCs, are derived from the earliest α-lymphoid progenitor cells (αLPs, CXCR6[+] integrin α4β7-expressing CLPs)[2], which differentiate into common helper-like innate lymphoid progenitor (CHILP) cells[14]. CHILPs generate all ILCs including LTi cells but not NK cells. Downstream of CHILP, ILC progenitors (ILCP), characterized by expression of the transcription factor (TF) PLZF, lose the ability to generate LTi cells and give rise to all ILC1, ILC2, and ILC3 subsets[15]. RORγt (encoded by Rorc) drives differentiation of ILC3s from their precursor ILCPs[16, 17]. RORγt deletion causes a complete loss of ILC3s but not ILC1s or ILC2s. Of note, the cytokine receptor chain IL-7Rα (CD127) is constitutively expressed in CHILPs and all ILCs, and forms a heterodimer with the common γ-chain of IL-2R or thymic stromal lymphopoietin (TSLP) receptor to detect IL-7 and TSLP, respectively[14, 18]. However, how IL-7Rα signaling regulates the ILC development and/or maintenance still remains elusive.

Protein post-translational modifications (PTM) such as phosphorylation, glycosylation, acetylation, and ubiquitination have critical functions in the regulation of activities of target proteins by changing their chemical or structural properties[19, 20]. Another PTM, glutamylation, adds glutamate side chains onto the γ-carboxyl groups of glutamic acid residues in the sequence of target proteins[21–23]. Glutamylation is catalyzed by polyglutamylases, also called tubulin tyrosine ligase-like (TTLL) enzymes[24, 25]. Glutamylation is a reversible modification that can be hydrolyzed by a family of cytosolic carboxypeptidases (CCPs)[26]. Misregulations of glutamylation contribute to several physiological abnormalities. CCP1 deficiency causes hyperglutamylation of tubulins resulting in Purkinje cell degeneration[26, 27]. We previously demonstrated that CCP6 deficiency induces hyperglutamylation of Mad2, leading to underdevelopment of megakaryocytes and abnormal thrombocytosis[28]. In addition, we also show that glutamylation of the DNA sensor cGAS regulates its binding and synthase activity in antiviral immunity[29], suggesting that glutamylation is involved in the regulation of immune response. However, how glutamylation regulates the development and/or maintenance of ILCs is unknown.

Here, we show that IL-7Rα can be glutamylated by TTLL4 and TTLL13, and deglutamylated by CCP2. IL-7Rα glutamylation enhances STAT5 activation and then promotes Sall3 transcription in CHILPs that drives the development of ILC3s. Therefore, IL-7Rα glutamylation has a critical function in ILC3 development.

## Results

**CCP2 deficiency increases ILC3 numbers**. We previously demonstrated that deficiency in CCP5 or CCP6 leads to susceptibility to virus infection[29]. CCP5 and CCP6 are required for the activation of TF IRF3 and IFN induction. We, therefore, sought to explore whether glutamylation was involved in the development of ILCs and their defense against bacterial infection. We used previously established Ccp1-6 knockout (KO) mice and further validated deletion of these genes in mouse bone marrow (BM) (Supplementary Fig. 1a). We analyzed ILC3s (Lin[−]CD45[+]RORγt[+]) in the small intestine lamina propria in all six deficient mouse strains and found that the number of ILC3 cells was significantly increased in Ccp2[−/−] mice, but not in other CCP KO mouse strains (Fig. 1a and Supplementary Fig. 1b, c). ILC3 cells can be divided into a set of subpopulations according to their expression of CD4 and NKp46 (encoded by Ncr1) receptors, such as CD4[+] ILC3s, NKp46[+] ILC3s, and CD4[−]NKp46[−]ILC3s (DN ILC3s)[6, 30]. We then determined changes of NKp46[+] ILC3s (Lin[−]CD45[+]RORγt[+]NKp46[+]) and NKp46[−] ILC3s (Lin[−]CD45[+]RORγt[+]NKp46[−]) in Ccp2[−/−] mice. We observed that both of NKp46[+] ILC3s and NKp46[−] ILC3s were markedly increased in Ccp2[−/−] mice, but not in other CCP-deficient mouse strains (Fig. 1b and Supplementary Fig. 1d). These observations were further verified by immunofluorescence staining (Fig. 1c). By contrast, CCP2-deficient mice displayed reduced numbers of ILC1s and ILC2s (Supplementary Fig. 1e, f).

NKp46[+] ILC3s substantially secrete IL-22[13, 31], which has a crucial function in the early host defense against Citrobacter (C.) rodentium infection. As expected, IL-22 secreting (IL-22[+]) ILC3s were three times increased in the small intestine of Ccp2[−/−] mice compared to that of littermate wild-type (WT) mice (Fig. 1d and Supplementary Fig. 1g). However, these IL-22[+] ILC3s were unchangeable in other CCP-deficient mouse strains (hereafter we used Ccp6 KO mice as a negative control) (Fig. 1d). We next infected Ccp2[−/−] or Ccp6[−/−] mice with C. rodentium. We noticed that Ccp2[−/−] mice were more resistant to C. rodentium infection compared with their littermate WT mice (Fig. 1e–g). By contrast, Ccp6[−/−] mice had comparable bacterial loads compared to their littermate WT mice. Furthermore, Ccp2[−/−] mice displayed increased numbers of IL-22[+] ILC3s in the small intestine after C. rodentium infection (Fig. 1h). In addition, higher expression of Il22 messenger RNA (mRNAs) in Ccp2[−/−] ILC3s was further confirmed with C. rodentium challenge (Fig. 1i). Consistently, with IL-23 stimulation, Ccp2[−/−] ILC3s produced much higher levels of IL-22 protein than WT ILC3s (Fig. 1j). By contrast, Ccp6[−/−] mice had no such effect. Taken together, CCP2 deficiency causes an increased number of ILC3s that enhance clearance of C. rodentium.

**CCP2 deficiency potentiates ILC3 differentiation from CHILPs**. We next analyzed expression patterns of CCP members in the mouse hematopoietic system. We found that Ccps displayed distinct expression profiles in different hematopoietic cell populations and their progenitors (Fig. 2a). Of note, Ccp2 was highly expressed in the CHILPs and ILC3s (Fig. 2a). Intriguingly, CCP2 deficiency led to reduced numbers of CHILPs, whereas more ILCPs in BM (Fig. 2b and Supplementary Fig. 1h), suggesting CCP2 was involved in the development of ILC3s from the stage of CHILPs. We then conducted in vitro differentiation assays. We isolated CHILPs from Ccp2[+/+] and Ccp2[−/−] mice and cultured them with OP9 feeder cells in the presence of murine IL-7 (25 ng/ml, Peprotech) and SCF (25 ng/ml, Peprotech). We noticed that Ccp2[−/−] CHILPs generated more ILC3s compared to Ccp2[+/+] CHILPs (Fig. 2c, d and Supplementary Fig. 1i). Moreover, overexpression of CCP2 dramatically reduced the formation of ILC3s, indicating that CCP2 was implicated in the development of ILC3. CoCl2 is an agonist for CCP family proteins[32], and phenanthroline (Phen) is their pan inhibitor[26].

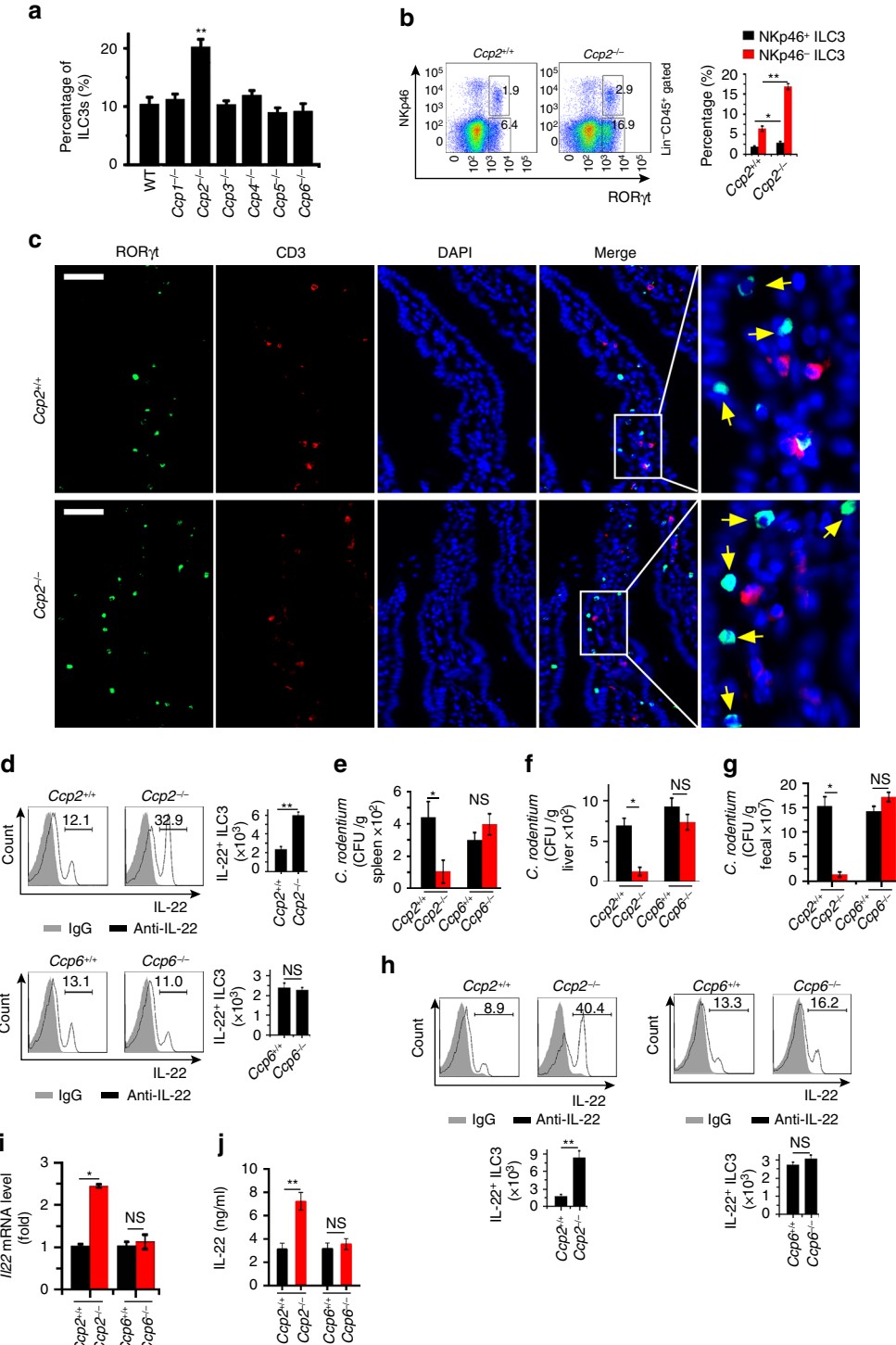

**Fig. 1** CCP2 deficiency increases ILC3 numbers. **a** Percentages of ILC3s (Lin⁻CD45⁺RORγt⁺) in small intestine lamina propria from wild-type (WT), *Ccp1⁻/⁻*, *Ccp2⁻/⁻*, *Ccp3⁻/⁻*, *Ccp4⁻/⁻*, *Ccp5⁻/⁻*, and *Ccp6⁻/⁻* mice were analyzed by flow cytometry. $n = 6$ for each group. **b** Analysis of NKp46⁺ ILC3s (Lin⁻CD45⁺RORγt⁺NKp46⁺) and NKp46⁻ ILC3s (Lin⁻CD45⁺RORγt⁺NKp46⁺) from *Ccp2⁺/⁺* and *Ccp2⁻/⁻* mice by flow cytometry. $n = 6$ for each group. **c** Analysis of ILC3s in *Ccp2⁺/⁺* and *Ccp2⁻/⁻* small intestines by immunofluorescence staining. *Arrowhead* denotes ILC3 cells. *Scale bars*, 50 μm. **d** Analysis of IL-22⁺ ILC3 in WT, *Ccp2⁻/⁻*, and *Ccp6⁻/⁻* small intestines after IL-23 stimulation. Cells were gated on Lin⁻ IL-22⁺. $n = 6$ for each group. **e–g** *C. rodentium* titers in spleen **e**, liver **f**, and fecals **g** from WT, *Ccp2⁻/⁻* and *Ccp6⁻/⁻* mice 8 day after infection. **h** Analysis of IL-22⁺ ILC3 in WT, *Ccp2⁻/⁻* and *Ccp6⁻/⁻* small intestines after *C. rodentium* infection. $n = 6$ for each group. **i** *Il22* expression was detected by real-time qPCR after *C. rodentium* infection. $n = 6$ for each group. **j** $1 \times 10^4$ ILC3s (Lin⁻CD45ˡᵒCD90ʰⁱ) isolated from WT, *Ccp2⁻/⁻* and *Ccp6⁻/⁻* intestines were cultured at 37 °C in vitro for 24 h in the presence of IL-23. IL-22 was examined by ELISA. $n = 6$ per group. *$P < 0.05$, **$P < 0.01$ (Student's *t*-test). NS, no significant. Data are representative of three independent experiments. *Error bars* in **a**, **b**, and **d–j** indicate s.d.

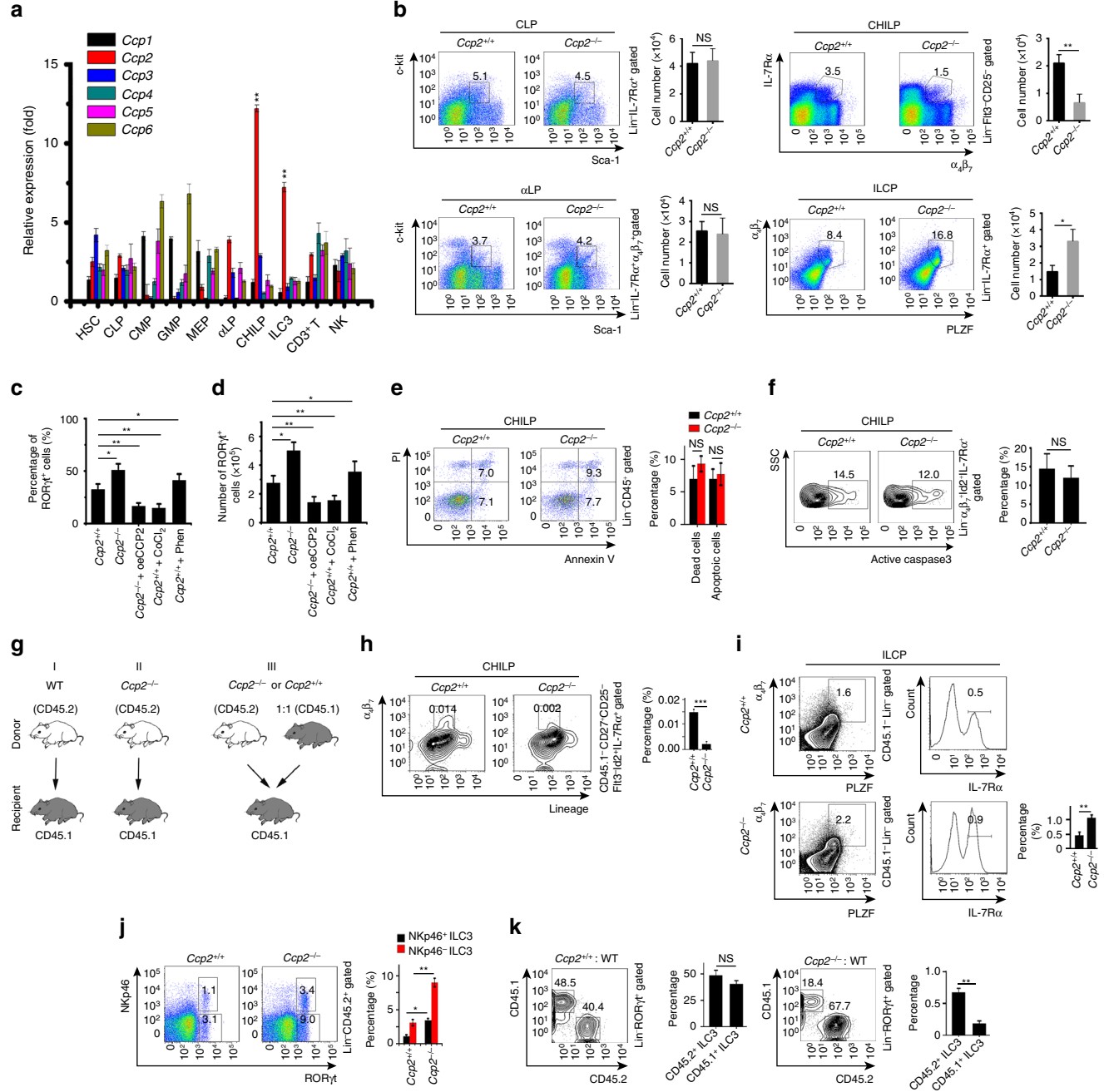

**Fig. 2** CCP2 deficiency potentiates ILC3 differentiation from CHILPs. **a** Total RNAs were extracted from hematopoietic populations. Indicated gene expression levels of *Ccp1*, *Ccp2*, *Ccp3*, *Ccp4*, *Ccp5*, and *Ccp6* were examined by real-time qPCR. **b** Gating strategies and flow cytometry analysis of CLP (Lin⁻IL-7Rα⁺Sca-1^low^c-Kit^low^), αLP (Lin⁻IL-7Rα⁺Sca-1⁺c-Kit⁺α₄β₇⁺), CHILP (Lin⁻IL-7Rα⁺Flt3⁻CD25⁻α₄β₇⁺), and ILCP (Lin⁻IL-7Rα⁺α₄β₇⁺PLZF⁺) in BM from *Ccp2*^+/+^ and *Ccp2*^−/−^ mice. $n = 6$ per group. Numbers of indicated cells in *Ccp2*^+/+^ and *Ccp2*^−/−^ mice were calculated as shown means±s.d. **c**, **d** CHILPs were isolated from *Ccp2*^+/+^ or *Ccp2*^−/−^ mice and cultured on OP9 cells in the presence of SCF and IL-7 for 12 days. Percentages **c** and total numbers **d** of ILC3s were examined by flow cytometry, gated on CD45⁺Lin⁻RORγt⁺. The CCP agonist CoCl₂ (10 μm) and antagonist Phen (2 μm) were added for in vitro differentiation assays. **e** Apoptosis of CHILPs from *Ccp2*^+/+^ and *Ccp2*^−/−^ mice was analyzed with Annexin V/PI. Apoptotic cells were calculated and shown as means±s.d. **f** Active caspase 3 in CHILPs from *Ccp2*^+/+^ and *Ccp2*^−/−^ mice was detected by flow cytometry. **g** Schematic representation for BM transplantation assays. **h–j** $5 \times 10^4$ CD45.2⁺ LSK from *Ccp2*^+/+^ or *Ccp2*^−/−^ mice with $5 \times 10^6$ CD45.1⁺ helper cells were transplanted into lethally irradiated CD45.1⁺ recipients. After 8 weeks, percentages of CHILPs **h**, ILCP **i**, and ILC3s **j** in chimeras were checked by FACS. $n = 6$ for each group. **k** A 50/50 mixture of CD45.1⁺ wild-type and CD45.2⁺ *Ccp2*^+/+^ or *Ccp2*^−/−^ bone marrow was transplanted into lethally irradiated CD45.1⁺ recipients. The ratio of CD45.1⁺ to CD45.2⁺ ILC3s in chimeras ($n = 6$) were analyzed by gating on CD45.2⁺Lin⁻RORγt⁺ (*Ccp2*^+/+^ or *Ccp2*^−/−^) and CD45.1⁺Lin⁻RORγt⁺ (WT). $*P < 0.05$, $**P < 0.01$ (Student's *t*-test). Data are representative of three independent experiments. *Error bars* in **a–f** and **h–k** indicate s.d.

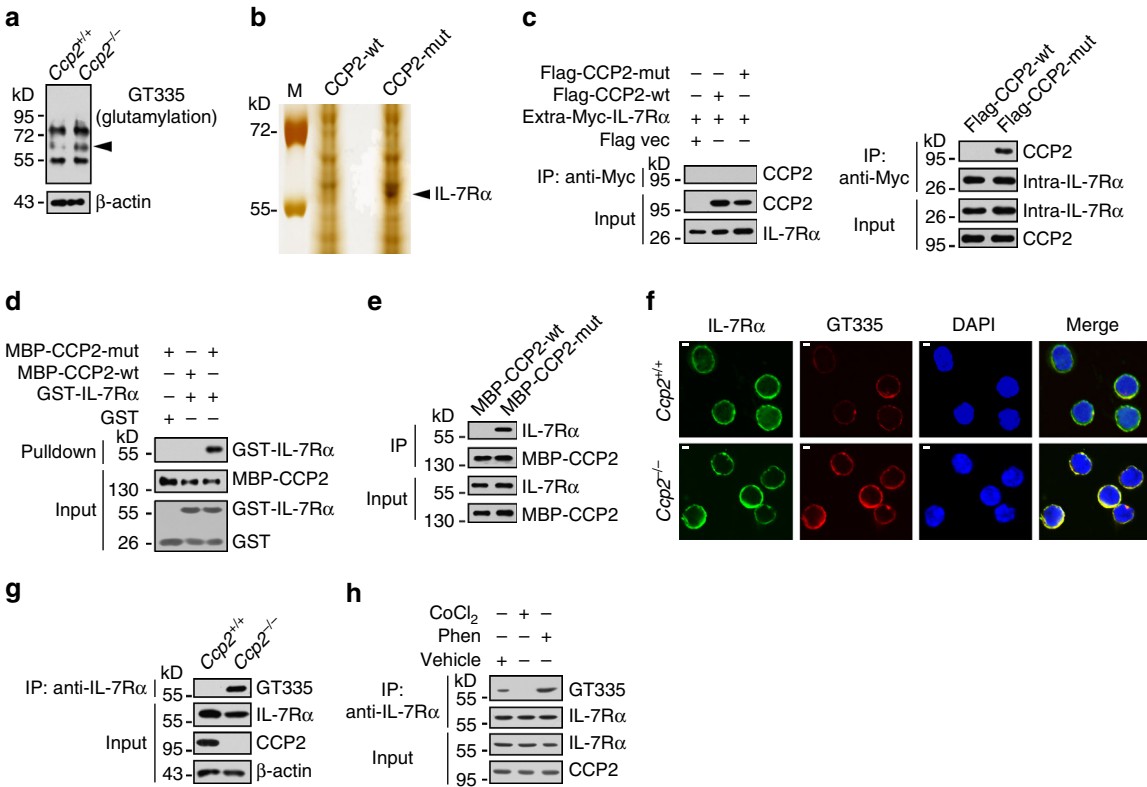

**Fig. 3** IL-7Rα is a substrate of CCP2 in CHILPs. **a** BM cells from WT and $Ccp2^{-/-}$ mice were lysed. Protein glutamylation was examined with GT335 antibody by western blotting. *Arrowhead* denotes the differential band. **b** Recombinant CCP2-wt and enzymatic inactive CCP2 mutant (CCP2-mut) were immobilized with Affi-gel10 resin and assessed by addition of $Ccp2^{-/-}$ BM lysates. The eluted fractions were resolved by SDS-PAGE, followed by silver staining. M: molecular weight marker. A differential band of ~60 kD appeared in CCP2-mut lane and was cut for mass spectrometry. The peptide sequences and coverage of IL-7Rα analyzed by LC-LTQ MS/MS are shown in the *bottom* graph. **c** Myc-tagged extracellular (amino acid: 21–239) or intracellular (amino acid: 265–459) segment of IL-7Rα and Flag-tagged CCP2-wt or CCP2-mut were co-transfected in 293 T cells for 36 h. Cell lysates were incubated with anti-Myc antibody for immunoprecipitation assay. *IP* immunoprecipitation. **d** GST-tagged intracellular segment of IL-7Rα (GST-IL-7Rα) was incubated with MBP-tagged CCP2-wt or CCP2-mut at 4 °C for 4 h, followed by incubation with GST beads. **e** CCP2-wt and CCP2-mut were incubated with BM lysates for pulldown assay. **f** CHILPs were incubated with GT335 and anti-IL-7Rα antibodies for immunofluorescence staining. IL-7Rα, *green*; GT335, *red*; nucleus, *blue*. Scale bar, 2 μm. **g** BM lysates from WT or $Ccp2^{-/-}$ mice were immunoprecipitated with anti-IL-7Rα antibody, followed by immunoblotting. **h** WT BM cells were treated with CoCl$_2$ or Phen. Cells were lysed and IL-7Rα glutamylation was assessed. Data represent four independent experiments

As expected, CoCl$_2$ treatment suppressed the generation of ILC3s, whereas Phen treatment increased the formation of ILC3s (Fig. 2c, d). Finally, $Ccp2^{-/-}$ CHILPs did not undergo apparent apoptosis (Fig. 2e, f). Altogether, polyglutamylation is required for the differentiation of ILC3s from their progenitor CHILPs.

**Cell-intrinsic modulations of ILC3 differentiation by CCP2.** We next sought to determine whether CCP2 deficiency-mediated ILC3 development was intrinsic or extrinsic. We transplanted CD45.2$^+$ $Ccp2^{-/-}$ or $Ccp2^{+/+}$ BM cells into lethally irradiated CD45.1$^+$ recipients (Fig. 2g). Eight weeks after transplantation, $Ccp2^{-/-}$ BM transferred recipients displayed reduced number of CHILPs but increased counts of ILCPs compared with those of $Ccp2^{+/+}$ BM transplantation (Fig. 2h, i). ILC3s were consequently increased in $Ccp2^{-/-}$ BM transferred recipients (Fig. 2j). We also performed competitive BM transplantation assays. We transferred a 1:1 mixture of CD45.1$^+$ WT and CD45.2$^+$ $Ccp2^{+/+}$ or $Ccp2^{-/-}$ BM into lethally irradiated recipient mice (Fig. 2g). Eight weeks after transplantation, reconstituted recipients showed increased numbers of ILC3s (Fig. 2k). Collectively, CCP2 is an intrinsic factor in the regulation of ILC3 development.

**IL-7Rα is a substrate of CCP2 in CHILPs.** To further explore the molecular mechanism of CCP2-mediated ILC3

differentiation, we analyzed lysates of $Ccp2^{+/+}$ and $Ccp2^{-/-}$ BM by immunoblotting with a glutamylation-specific antibody GT335. The antibody GT335 specifically recognizes the branch points of glutamate side chains and detects all glutamylation forms of target proteins[26]. After immunoblot analysis, one band around 60 kD appeared in the lane of CCP2-deficient BM lysates (Fig. 3a). This band was undetectable in the corresponding lane location from the littermate control BM lysates. Thus, this band could be a potential candidate substrate for CCP2. To identify the candidate substrates of CCP2, we generated an enzymatically inactive mutant of CCP2 (CCP2-mut) through H425S and E428Q mutations as previously described[26]. WT CCP2 (CCP2-wt) and CCP2-mut were immobilized with Affi-gel10 resin to go through mouse BM lysates for affinity chromatography. The eluted fractions were resolved by sodium dodecyl sulfate polyacrylamide gel electrophoresis (SDS-PAGE), followed by silver staining. This band was present in the gel analyzing CCP2-mut and was cut for mass spectrometry, whose band was identified as IL-7Rα (Fig. 3b and Supplementary Fig. 2a).

We transfected Myc-tagged intracellular (amino acid: 265–459) or Myc-tagged extracellular segment (amino acid: 21–239) of IL-7Rα with Flag-tagged CCP2-wt or CCP2-mut into 293 T cells for co-immunoprecipitation assay. We found that the Flag-tagged enzymatic dead CCP2 (Flag-CCP2-mut) could pull down the Myc-tagged intracellular segment of IL-7Rα (hereafter we called

IL-7Rα) (Fig. 3c). By contrast, Flag-CCP2-mut failed to precipitate Myc-tagged extracellular segment of IL-7Rα (Fig. 3c). Consistently, glutamylated GST-tagged intracellular segment of IL-7Rα protein could pull down MBP-CCP2-mut by a pulldown assay (Fig. 3d), suggesting the intracellular segment of IL-7Rα was deglutamylated by CCP2. Moreover, MBP-tagged mutant CCP2 (MBP-CCP2-mut) was able to pull down native IL-7Rα from BM lysates, whereas the enzymatic active CCP2 (MBP-CCP2-wt)

could not precipitate IL-7Rα (Fig. 3e). These observations suggest that the intracellular segment of IL-7Rα undergoes deglutamylation by CCP2. With immunofluorescence staining, hyperglutamylation of IL-7Rα appeared in primary CCP2-deficient CHILPs (Fig. 3f). In parallel, IL-7Rα was highly polyglutamylated in BM lysates of $Ccp2^{-/-}$ mice (Fig. 3g). Finally, BM cells treated with the CCP antagonist Phen increased substantial amounts of polyglutamylated IL-7Rα, whereas treatment with the CCP

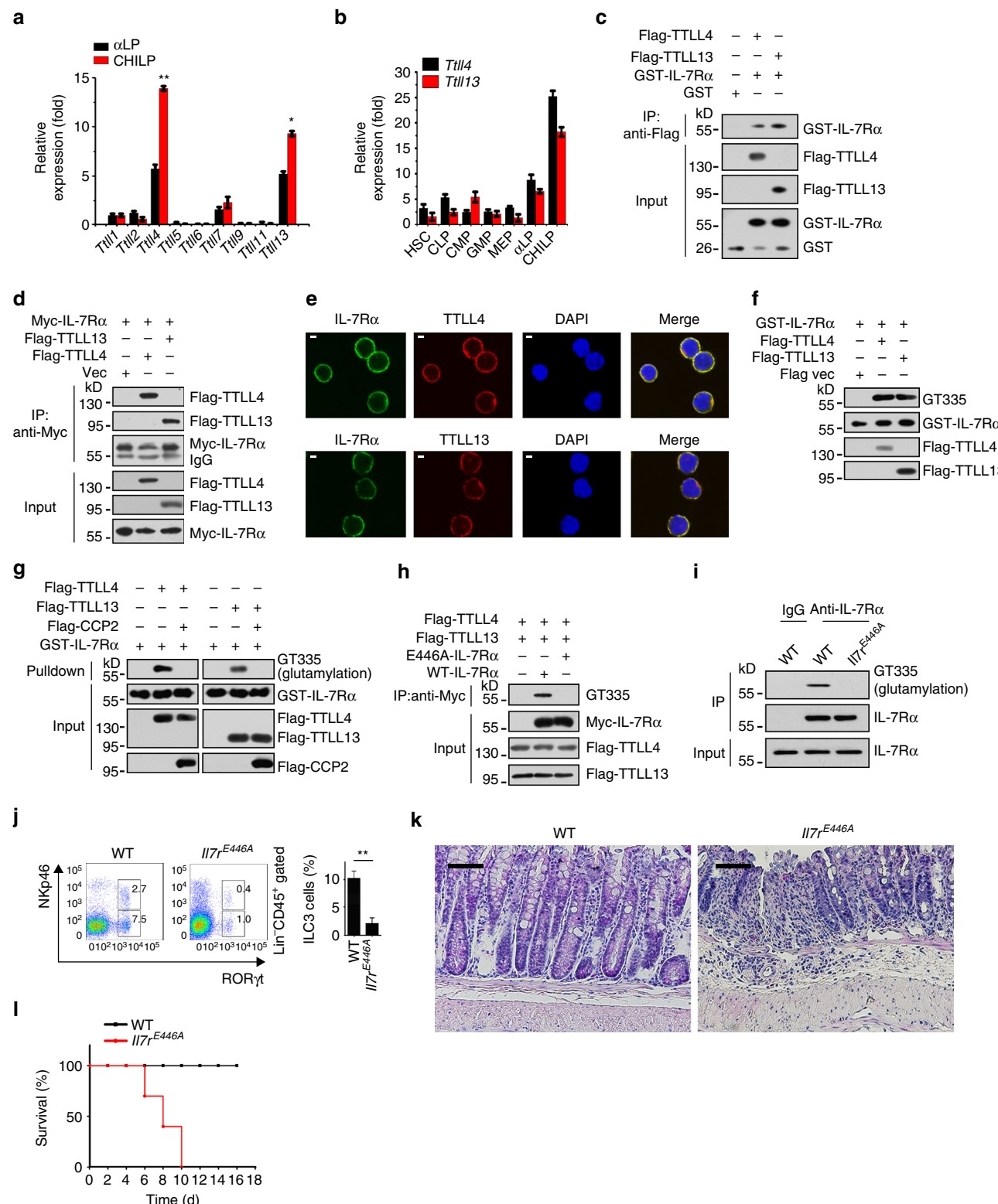

agonist $CoCl_2$ abrogated the glutamylation of IL-7Rα (Fig. 3h). Collectively, we conclude that IL-7Rα is a novel substrate for CCP2.

**IL-7Rα is polyglutamylated at Glu446 by TTLL4 and TTLL13.**
Nine polyglutamylases have been reported to catalyze protein glutamylation[21, 24]. To determine the physiological polyglutamylases catalyzing IL-7Rα glutamylation, we examined expression patterns of all nine polyglutamylases in αLPs and CHILPs of mouse BM through quantitative real-time PCR. We observed that Ttll4 and Ttll13 were highly expressed in αLPs and CHILPs, with peak expression in CHILPs (Fig. 4a). Additionally, Ttll4 and Ttll13 were highest expressed in CHILPs among all the hematopoietic progenitor cells (Fig. 4b). We next incubated recombinant intracellular segment rGST-IL-7Rα with Flag-tagged TTLL4 or TTLL13 in vitro. We noticed that Flag-tagged TTLL4 and TTLL13 were able to precipitate rGST-IL-7Rα (Fig. 4c). Their interactions were further verified by co-transfection assays (Fig. 4d). Moreover, IL-7Rα was co-localized with TTLL4 and TTLL13 in CHILPs (Fig. 4e). We then conducted in vitro glutamylation assays by incubation of rGST-IL-7Rα with Flag-TTLL4 or Flag-TTLL13. We found that rGST-IL-7Rα was polyglutamylated by TTLL4 and TTLL13 (Fig. 4f). Importantly, TTLL4- and TTLL13-mediated polyglutamylation of rGST-IL-7Rα was successfully removed by enzymatic active CCP2 (Fig. 4g). These data indicate that TTLL4 and TTLL13 are two polyglutamylases for IL-7Rα polyglutamylation.

Glutamate-rich stretches and acidic environment at the acceptor sites have been reported to be important for glutamylation modification[33]. Based on the conservative amino-acid sequence analysis, only Glu446 and Glu447 were two conserved identical glutamic acid residues located on the loop region of intracellular domain of IL-7Rα (Supplementary Fig. 2b), which might be potential acceptor site candidates for glutamylation. We then mutated Glu446 to Ala of IL-7Rα (E446A-IL-7Rα) and incubated recombinant intracellular E446A-IL-7Rα protein with Flag-TTLL4 or Flag-TTLL13 in vitro. We observed that E446A-IL-7Rα mutant abolished TTLL4- or TTLL13-mediated glutamylation (Fig. 4h), suggesting IL-7Rα is catalyzed by TTLL4 and TTLL13 at Glu446.

We next explored the physiological relevance of IL-7Rα glutamylation in ILC3 differentiation. We silenced IL-7Rα by LMP retrovirus-carried short hairpin RNA (shRNA) infection in CHILPs and then rescued expression of WT-IL-7Rα or E446A-IL-7Rα, followed by BM transplantation assays. Eight weeks after transplantation, IL-7Rα knockdown with empty vector infection remarkably decreased ILC3 numbers (Supplementary Fig. 2c). By contrast, WT-IL-7Rα restoration could rescue the normal number of ILC3s in recipient mice, whereas E446A-IL-7Rα mutant overexpression had no such effect (Supplementary Fig. 2c). Parallelly, these observations were further validated by in vitro differentiation assays (Supplementary Fig. 2d, e).

Collectively, IL-7Rα glutamylation is required for ILC3 development.

To further validate the authentic role of IL-7Rα glutamylation in the regulation of ILC3 development, we generated E446A-IL-7Rα mutation ($Il7r^{E446A}$) mice through CRISPR/Cas9 technology. We noticed that IL-7Rα really did not undergo glutamylation in CHILPs of $Il7r^{E446A}$ mice (Fig. 4i). Consistently, $Il7r^{E446A}$ mice displayed reduced numbers of ILC3s (Fig. 4j and Supplementary Fig. 2f) and more severe intestinal injury post C. rodentium infection (Fig. 4k). Consequently, $Il7r^{E446A}$ mice died rapidly with C. rodentium challenge (Fig. 4l). Finally, we transplanted CD45.2$^+$ WT or $Il7r^{E446A}$ BM cells into lethally irradiated CD45.1$^+$ recipients. Eight weeks after transplantation, $Il7r^{E446A}$ BM transferred recipients displayed a reduced number of ILC3s compared to those of WT BM engraftment (Supplementary Fig. 2g). Collectively, we conclude that IL-7Rα glutamylation is required for ILC3 development from CHILPs.

**IL-7Rα glutamylation promotes Sall3 expression by STAT5.**
IL-7Rα pairs with the common γ-chain of IL-2R or TSLP receptor to detect IL-7 and TSLP, respectively, for the activation of STAT proteins in DCs, CD4$^+$ T as well as B cells[18, 34, 35]. However, how IL-7Rα glutamylation regulates the development of ILCs remains unclear. We then analyzed all STAT protein phosphorylation signals in $Ccp2^{+/+}$ and $Ccp2^{-/-}$ CHILPs. We found that only STAT5 was hyperphosphorylated in $Ccp2^{-/-}$ CHILPs compared to $Ccp2^{+/+}$ CHILPs with IL-7 stimulation (Fig. 5a). However, other STAT proteins were not activated (Fig. 5a). We thus used STAT3 as a negative control in the following experiments. These observations were further validated by flow cytometry and immunofluorescence staining (Fig. 5b, c). These results indicate that IL-7Rα glutamylation leads to STAT5 activation in CHILPs after IL-7 stimulation.

To further determine which TFs regulated IL-7Rα glutamylation-mediated ILC3 development, we performed transcriptome microarray analysis of $Ccp2^{+/+}$ vs. $Ccp2^{-/-}$ CHILPs. Among top 10 upregulated TFs in $Ccp2^{-/-}$ CHILPs (Fig. 5d), we focused on Spalt-like transcription factor 3 (Sall3), which was a highest differentially expressed TF in $Ccp2^{-/-}$ CHILPs (Fig. 5e). Sall3, a member of the SAL family, is implicated in embryonic development and oncogenesis[36, 37]. However, how Sall3 regulates the development of ILC3s is unknown. Intriguingly, the promoter region of Sall3 gene contained the STAT5-binding motif (TTCNNNGAA)[38] (Supplementary Fig. 2h). Of note, anti-STAT5 antibody could immunoprecipitate Sall3 promoter by chromatin immunoprecipitation (ChIP) assay, whereas the anti-STAT5 antibody failed to immunoprecipitate other TF promoters (Fig. 5f). Consistently, the Sall3 promoter of $Ccp2^{-/-}$ CHILPs bound to substantial STAT5 proteins compared to that of $Ccp2^{+/+}$ CHILPs (Fig. 5g). The binding of STAT5 to Sall3 promoter was further verified by an EMSA assay (Fig. 5h). Consequently, STAT5 was able to activate Sall3 transcription

**Fig. 4** IL-7Rα is polyglutamylated at Glu446 by TTLL4 and TTLL13. **a** Gene expression levels of Ttll1, Ttll2, Ttll4, Ttll5, Ttll6, Ttll7, Ttll9, Ttll11, and Ttll13 were examined in αLPs and CHILPs by real-time qPCR. Relative fold changes of gene expression values were normalized to endogenous Actb. **b** Ttll4 and Ttll13 expression patterns were analyzed in hematopoietic populations as in **a**. **c** rGST-IL-7Rα bound to Flag-TTLL4 and Flag-TTLL13 by pulldown assay. **d** Myc-IL-7Rα with Flag-TTLL4 or Flag-TTLL13 were co-transfected in 293T cells for 36 h, followed by immunoblotting assay. **e** CHILPs were incubated with antibodies against TTLL4, TTLL13, and IL-7Rα by immunofluorescence staining. IL-7Rα, green; TTLL4 and TTLL13, red; nucleus, blue. Scale bar, 2 μm. **f** rGST-IL-7Rα was incubated with TTLL4 and TTLL13 for in vitro glutamylation assay. **g** TTLL4- and TTLL13-mediated rGST-IL-7Rα glutamylation was removed by CCP2. **h** Flag-TTLL4 and Flag-TTLL13 with Myc-IL-7Rα-wt or Myc-IL-7Rα-E446A were transfected into 293T cells. Cell lysates were immunoprecipitated by anti-Myc antibody and immunoblotted with indicated antibodies. **i** IL-7Rα glutamylation was analyzed in BM cells from WT and $Il7r^{E446A}$ by immunoblotting. **j** ILC3s were analyzed in WT and $Il7r^{E446A}$ mice by FACS. n = 6 for each group. **k** Histology of colons from WT and $Il7r^{E446A}$ mice 8 days after infection with C. rodentium. Scale bars, 50 μm. **l** Survival rates were monitored at the indicated time points with C. rodentium infection. n = 8 for each group. **P < 0.01 (Student's t-test). Data are representative of three independent experiments. Error bars in **a**, **b**, and **j** indicate s.d.

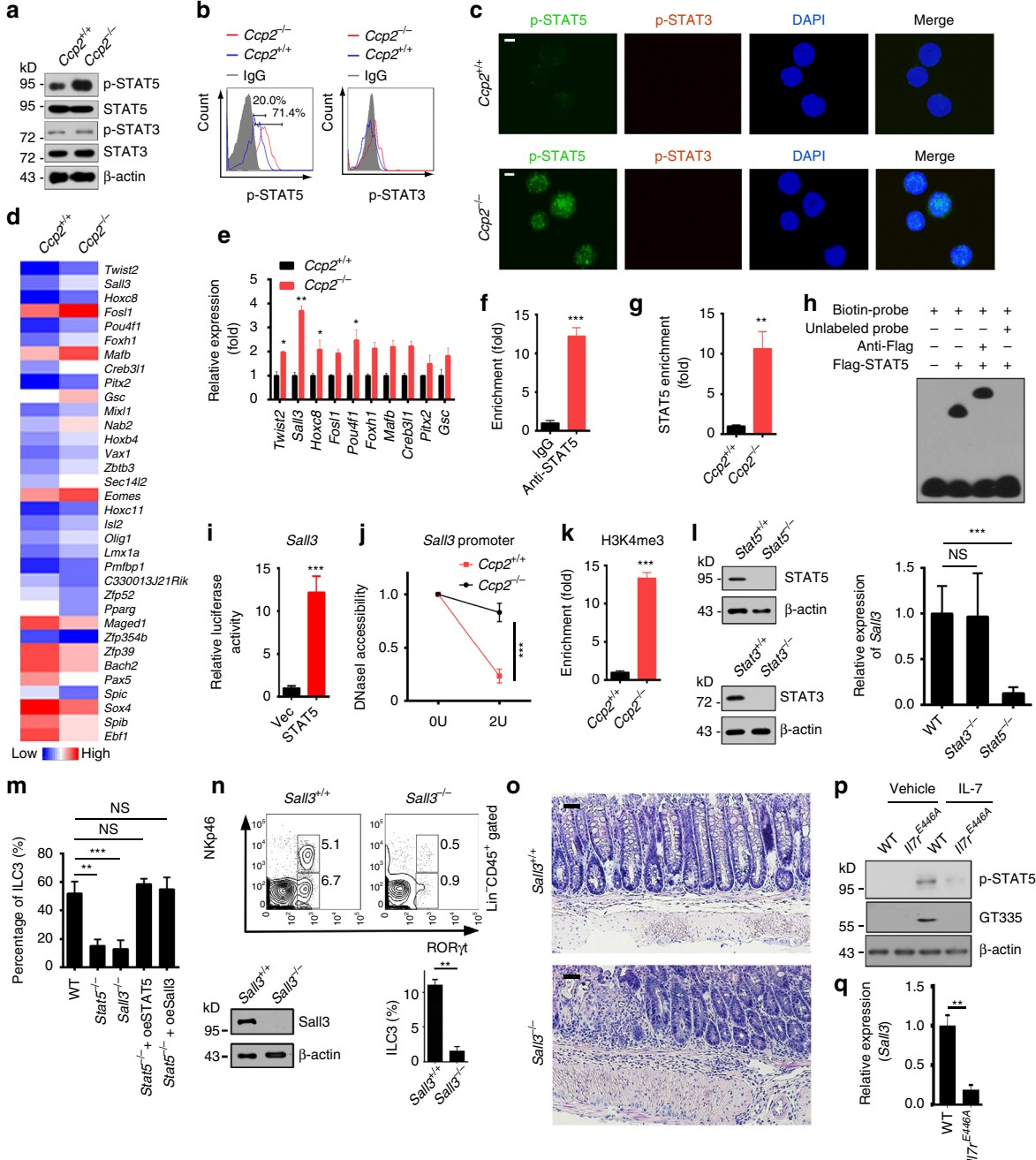

**Fig. 5** IL-7Rα glutamylation promotes *Sall3* expression by STAT5. **a–c** Phosphorylation of STAT3 and STAT5 was tested by western blotting **a**, flow cytometry **b** and immunofluorescence staining **c** in CHILPs from WT and *Ccp2⁻/⁻* mice after IL-7 stimulation. p-STAT5, *green*; p-STAT3, *red*; nucleus, *blue*. **d** Heat map of representative gene expression values from microarray data. In all, 1 × 10⁶ CHILPs (Lin⁻Flt3⁻CD25⁻IL-7Rα⁺α₄β₇⁺) from *Ccp2⁺/⁺* or *Ccp2⁻/⁻* mice were sorted for microarray. **e** Analysis of indicated gene expression levels in *Ccp2⁺/⁺* and *Ccp2⁻/⁻* CHILPs by quantitative reverse transcription PCR (RT-qPCR). Relative fold changes of gene expression values were normalized to endogenous *Actb*. **f, g** Enrichment assessment of STAT5 on *Sall3* promoter in CHILPs from *Ccp2⁺/⁺* and *Ccp2⁻/⁻* mice. **h** The association of STAT5 with *Sall3* promoter was examined by EMSA. *Sall3* promoter probe was biotin-labeled. **i** Flag-STAT5, pTK, and pGL3- *Sall3* promoter were transfected into 293T cells for luciferase assay. **j** DNaseI accessibility of *Sall3* promoter in CHILPs from *Ccp2⁺/⁺* and *Ccp2⁻/⁻* mice was assessed. **k** H3K4me3 enrichment on *Sall3* promoter was determined. CHILPs were isolated from *Ccp2⁺/⁺* and *Ccp2⁻/⁻* mice, followed by ChIP assay. **l** *Sall3* mRNA levels were detected in WT, *Stat3⁻/⁻*, and *Stat5⁻/⁻* CHILPs. **m** CHILPs were isolated from WT mice and cultured with mitomycin C-treated OP9 cells for in vitro differentiation assay. *Stat5⁻/⁻* CHILPs were transfected with STAT5 or Sall3 overexpression plasmid. Percentages of ILC3s were analyzed by flow cytometry. *oe* overexpression. **n** ILC3s were analyzed in *Sall3⁺/⁺* and *Sall3⁻/⁻* mice. *n* = 6 for each group. **o** Histology of colons from *Sall3⁺/⁺* and *Sall3⁻/⁻* mice 8 days after *C. rodentium* infection. *Scale bars*, 50 μm. **p** IL-7Rα glutamylation and STAT5 phosphorylation in CHILPs in the absence or presence of IL-7. CHILPs from WT or *Il7rᴱ⁴⁴⁶ᴬ* BM were incubated with IL-7 and vehicle, followed by immunoblotting. β-actin was probed as loading controls. **q** *Sall3* expression levels were tested in WT and *Il7rᴱ⁴⁴⁶ᴬ* CHILPs by RT-qPCR. Relative fold change of Sall3 expression values were normalized to endogenous *Actb*. **P < 0.01, ***P < 0.001 (Student's *t*-test). Data represent three independent experiments. *Error bars* in **e–g**, **i–n**, and **q** indicate s.d.

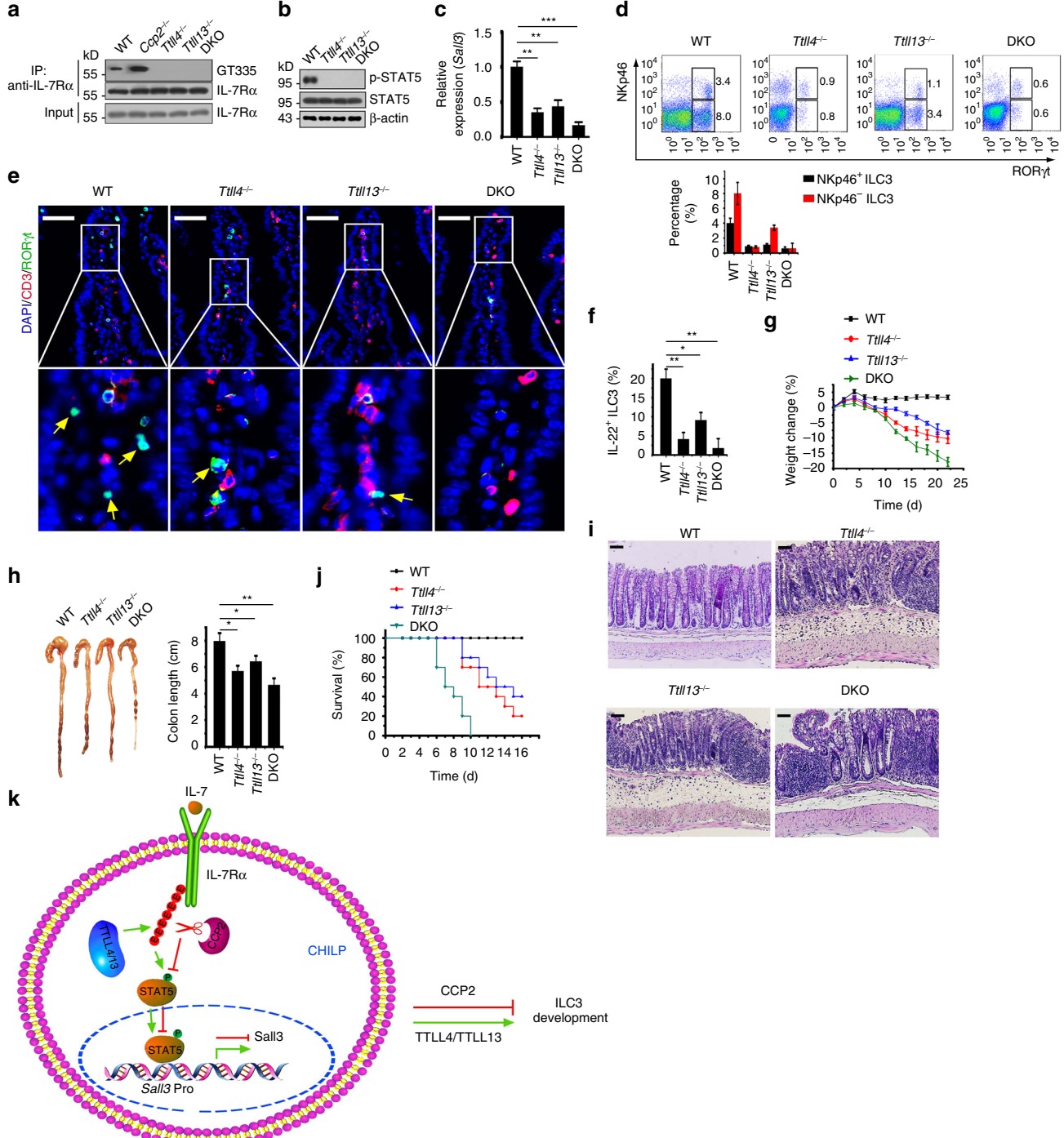

**Fig. 6** Deletion of *Ttll4* or *Ttll13* impairs ILC3 development. **a** Detection of IL-7Rα glutamylation in CHILPs by gating on Lin⁻IL-7Rα⁺Flt3⁻CD25⁻α₄β₇⁺ from WT, *Ccp2⁻/⁻*, *Ttll4⁻/⁻*, *Ttll13⁻/⁻*, and *Ttll4⁻/⁻Ttll13⁻/⁻* mice. **b** Examination of STAT5 phosphorylation in CHILPs from WT, *Ccp2⁻/⁻*, *Ttll4⁻/⁻*, *Ttll13⁻/⁻*. and *Ttll4⁻/⁻Ttll13⁻/⁻* mice. **c** *Sall3* expression levels were tested in WT, *Ttll4⁻/⁻*, *Ttll13⁻/⁻*, and *Ttl4⁻/⁻Ttll13⁻/⁻* CHILPs by quantitative reverse transcription PCR (RT-qPCR). Relative fold change of Sall3 expression values were normalized to endogenous *Actb*. **d** Flow cytometry analysis of ILC3s in small intestine lamina propria from WT, *Ttll4⁻/⁻*, *Ttll13⁻/⁻*, and *Ttll4⁻/⁻Ttll13⁻/⁻* mice. Percentages of indicated cells were calculated and shown as means±s.d. (*right* panel). n = 6 for each group. **e** Analysis of ILC3s in WT, *Ttll4⁻/⁻*, *Ttll13⁻/⁻*, and *Ttl4⁻/⁻Ttll13⁻/⁻* small intestines by immunofluorescence staining in situ. *Arrowhead* denotes ILC3 cells. *Scale bars*, 50 μm. **f** Analysis of IL-22⁺ ILC3s after IL23 stimulation by flow cytometry. n = 6 for each group. **g** Body weight changes of WT, *Ttll4⁻/⁻*, *Ttll13⁻/⁻*, and *Ttll4⁻/⁻Ttll13⁻/⁻* mice post *C. rodentium* infection. n = 10 for each group. **h** Colon length from WT, *Ttll4⁻/⁻*, *Ttll13⁻/⁻*, and *Ttll4⁻/⁻Ttll13⁻/⁻* mice after *C. rodentium* infection. n = 6 per genotype group. **i** Histology of colons from WT, *Ttll4⁻/⁻*, *Ttll13⁻/⁻*, and *Ttll4⁻/⁻Ttll13⁻/⁻* mice 8 days after infection with *C. rodentium*. *Scale bars*, 50 μm. **j** Survival curves of WT, *Ttll4⁻/⁻*, *Ttll13⁻/⁻*, and *Ttll4⁻/⁻Ttll13⁻/⁻* mice after *C. rodentium* infection. n = 10 for each group. In all, 2 × 10⁹ *C. rodentium* was used by oral inoculation. **k** A working model represents glutamylation-mediated IL-7 signaling in the regulation of ILC3 development. *Pro* promoter. \*P < 0.05, \*\*P < 0.01, and \*\*\*P < 0.001 (Student's *t*-test). Data represent three independent experiments. *Error bars* in **c**, **d**, and **f–h** indicate s.d.

through in vitro luciferase assays (Fig. 5i). Moreover, *Sall3* promoter in *Ccp2*[−/−] CHILPs was more accessible to DNase I digestion (Fig. 5j) and enriched more H3K4me3 (Fig. 5k), suggesting CCP2 deficiency promotes *Sall3* expression.

We next generated *Stat5* KO CHILPs via Cas9 knockin mice as described[39] and *Stat3* conditional KO mice by crossing *Stat3*[flox/flox] mice to *Mx1*-Cre mice. STAT5 and STAT3 were successfully deleted in CHILPs (Fig. 5l). We noticed that *Stat5*[−/−] CHILPs abrogated *Sall3* expression, while *Stat3*[−/−] CHILPs did not impact the expression of *Sall3* (Fig. 5l). We then cultured CHILPs with OP9 cells in the presence of SCF and IL-7 in vitro. We found that *Stat5* KO drastically reduced ILC3 numbers (Fig. 5m). However, rescue expression of STAT5 or Sall3 in *Stat5*[−/−] CHILPs was able to restore ILC3 numbers comparable to WT mice (Fig. 5m), suggesting Sall3 was a downstream target for STAT5 activation during ILC3 differentiation. We also generated Sall3 KO mice via Cas9 knockin mice. We observed that Sall3 KO mice also displayed decreased ILC3s and more susceptibility to *C. rodentium* infection compared with WT control mice (Fig. 5n, o and Supplementary Fig. 2i). We next engrafted $5 \times 10^4$ CD45.2[+] LSKs from WT or *Sall3*[−/−] mice with $5 \times 10^6$ CD45.1[+] helper cells into lethally irradiated CD45.1[+] recipients for 8 weeks, followed by analysis of mixed chimeras. We found that *Sall3*[−/−] BM transferred recipients displayed a reduced number of ILC3s compared to those of WT BM engraftment (Supplementary Fig. 2j). More importantly, IL-7Rα polyglutamylation indeed occurred in WT CHILPs with IL-7 stimulation, and STAT5 phosphorylation appeared as well (Fig. 5p). However, in the absence of IL-7, IL-7Rα did not undergo polyglutamylation and no STAT5 phosphorylation appeared in WT CHILPs (Fig. 5p). By contrast, in the presence of IL-7, CHILPs from *Il7r*[E446A] mice did not undergo polyglutamylation and no STAT5 phosphorylation appeared either (Fig. 5p). Finally, *Il7r*[E446A] CHILPs did not activate the expression of Sall3 (Fig. 5q). Taken together, we conclude that IL-7Rα glutamylation-mediated STAT5 activation initiates Sall3 expression that drives the development of ILC3s from CHILPs.

**Deletion of *Ttll4* or *Ttll13* impairs ILC3 development**. We next generated *Ttll4* and *Ttll13* KO mice via CRISPR/Cas9 technology (Supplementary Fig. 3a, b). We noticed that deletion of TTLL4 or TTLL13 abrogated IL-7Rα glutamylation in CHILPs (Fig. 6a). Importantly, *Ttll4*[−/−] or *Ttll13*[−/−] CHILPs impaired STAT5 phosphorylation and blocked Sall3 expression in CHILPs (Fig. 6b, c). Consistently, *Ttll4* or *Ttll13* KO abolished ILC3 differentiation (Fig. 6d and Supplementary Fig. 3c). *Ttll4* and *Ttll13* double KO (DKO) almost suppressed the ILC3 formation in small intestines (Fig. 6d). These results were further confirmed by immunofluorescence staining (Fig. 6e). Additionally, *Ttll4* or *Ttll13* KO also dramatically reduced numbers of IL-22[+] ILC3s (Fig. 6f). As expected, *Ttll4* or *Ttll13* KO remarkably declined ILCP generation (Supplementary Fig. 3d). These data suggest that TTLL4- and TTLL13-mediated IL-7Rα glutamylation is required for the activation of Sall3 in CHILPs that drives the differentiation of ILC3s.

We next infected WT, *Ttll4*[−/−], *Ttll13*[−/−], or *Ttll4*[−/−]*Ttll13*[−/−] mice with *C. rodentium*. We observed that *Ttll4*[−/−] and *Ttll13*[−/−] mice had higher bacterial loads in fecal, spleen, liver, and blood on day 8 post infection compared with their littermate WT control mice (Supplementary Fig. 3e–h). Additionally, *Ttll4*[−/−] and *Ttll13*[−/−] mice lost their weights over *C. rodentium* infection (Fig. 6g), accompanied with shrinking length of colons (Fig. 6h). Following infection with *C. rodentium*, *Ttll4*[−/−] and *Ttll13*[−/−] mice displayed much more persistent intestinal damage, encompassing greater epithelial injury, crypt hyperplasia, and more

infiltration of inflammatory cells, than those of their littermate WT control mice (Fig. 6i). Consistently, *Ttll4*[−/−] and *Ttll13*[−/−] mice succumbed to bacterial infection (Fig. 6j). Expectedly, *Ttll4* and *Ttll13* DKO displayed much higher susceptibility to *C. rodentium* infection (Fig. 6g–j). In sum, TTLL4- and TTLL13-mediated IL-7Rα glutamylation has a critical function in the differentiation and effector functions of ILC3s.

**Discussion**

ILCs are a distinct arm of the innate immune system, which can directly communicate with other hematopoietic and non-hematopoietic cells to regulate immunity, inflammation and tissue homeostasis[1]. However, how these ILC lineages develop and/or maintain remains unclear. In this study, we show that CCP2 deficiency causes increased numbers of ILC3s. With IL-7 engagement, IL-7Rα undergoes polyglutamylation in CHILPs. IL-7Rα polyglutamylation specifically activate STAT5 phosphorylation to initiate Sall3 expression for ILC3 development (Fig. 6k). In addition, *Ttll4*[−/−] and *Ttll13*[−/−] mice abrogate IL-7Rα polyglutamylation and Sall3 expression in CHILPs, leading to impaired ILC3 differentiation and more susceptibility to *C. rodentium* infection. Finally, E446A-IL-7Rα mutation mice indeed abrogates Sall3 expression and ILC3 development.

The earliest progenitor cells specific to ILCs are CXCR6[+] integrin $α_4β_7$-expressing CLPs, referred to as α-lymphoid precursor (αLP) cells, which give rise to ILC1, ILC2, ILC3, and conventional NK cells (cNK)[40]. The common progenitor to all ILC lineages (CHILP) is identified as its Lin[−]IL-7Rα[+] Id2[+]CD25[−]$α_4β_7$[+] phenotype and differentiates to all ILC subsets, but not cNKs[14]. The common precursor to ILCs (ILCP) is defined by expression of TF PLZF and generates ILC1, ILC2, and ILC3 subpopulations[15]. In this study, we show that CCP2 is highly constitutively expressed in CHILPs and ILC3s, which blocks the deglutamylation of IL-7Rα to drive ILC3 development. CCP2 deficiency causes increased numbers of ILC3s, but reduced numbers of ILC1s and ILC2s, which augments clearance of *C. rodentium*. Given that CCP2 is also moderately expressed in other cells such as CD3[+] T cells, we thus cannot exclude the potential involvement of other cells in the bacterial clearance of CCP2 deficiency. Of note, CCP2 deficiency does not impact cell deaths of CHILPs and all ILC lineages. A recent study showed that different ILC subsets are defined by distinct gene-expression patterns[41]. Of note, cytokines such as IL-7, IL-15, and IL-2 play major roles in the regulation of ILC development. However, how CCP2-mediated IL-7 signaling regulates the switch balance of ILC development still needs to be further investigated. We notice that CCP members are differentially expressed in the hematopoietic progenitors and lineages we checked. We previously demonstrated that CCP6 is mostly highly expressed in BM and megakaryocytes, and also exhibits different expression profiles in different tissues and cell types[28]. Our findings suggest that different tissue and cell type distributions of CCPs may exert unique roles in the modulation of different physiological and pathological processes.

Protein polyglutamylation is catalyzed by a family of poly-glutamylases, also called TTLLs[24, 25]. The well-known substrates of polyglutamylation are tubulins and nucleosome assembly proteins[33]. Through regulating the interaction of microtubules (MTs) and MT-associated proteins (MAPs), polyglutamylation may exert major effects on MT-related cellular processes, including stability of centrosomes[42], motility of cilia and flagella[43, 44], neurite outgrowth[45], as well as neurodegeneration[26]. A recent study delineates a structural MT recognition basis by catalysis with TTLL7[21]. TTLLs have different expression patterns in diverse tissues and their functions are not entirely redundant[43].

We recently reported that TTLL4 and TTLL6 are most highly expressed in megakaryocytes[28], both of whom catalyze poly-glutamylation of Mad2 to modulate megakaryocyte maturation. Here we demonstrate that TTLL4 and TTLL13 are constitutively elevated in CHILPs, both of which can catalyze polyglutamylation of IL-7Rα to regulate the development of ILC3s. Deletion of TTLL4 or TTLL13 impairs ILC3 differentiation and their effector functions. Thus, IL-7Rα polyglutamylation mediated by TTLL4 or TTLL13 has a critical function in the regulation of ILC3 development from the stage of CHILPs.

IL-7Rα (CD127), encoded by *Il7r* gene, forms a receptor complex with the common cytokine receptor γ-chain of IL-2R or TSLP receptor to sense IL-7 and TSLP, respectively[18, 46]. The IL-7–IL-7Rα ligand-receptor pair signaling is critical for proliferation and survival of T and B lymphocytes in a non-redundant fashion. Genetic aberrations of IL-7Rα signaling lead to immune deficiency syndromes and other immune diseases[47, 48]. It has been reported that all ILC lineages express high levels of IL-7Rα[14]. Of note, the ILCP CHILPs also express IL-7Rα, which gives rise to all ILCs. However, the molecular mechanism by which IL-7Rα signaling regulates the development of ILCs remains elusive. In this study, we show that TTLL4 and TTLL13-mediated IL-7Rα polyglutamylation regulates the differentiation of ILC3s from CHILPs. Mechanistically, polyglutamylated IL-7Rα is able to activate STAT5 and phosphorylated STAT5 can directly bind to *Sall3* promoter to initiate its transcription, which drives the development of ILC3s from CHILPs.

A CHILP cell has been defined that lacks expression of Flt3 and CD25 but expresses IL-7Rα and $α_4β_7$[14]. CHILPs differ from α-LPs in that CHILPs express Id2. CHILPs generate all ILCs, including LTi cells, but they fail to give rise to conventional NK cells. Subsequently, their downstream precursor ILCPs (common precursor of ILCs), characterized by expression of the TF PLZF, lose the ability to generate LTi cells and produce all ILC1, ILC2, and ILC3 subsets[15]. RORγt (encoded by *Rorc*) drives differentiation of ILC3s from their precursor ILCPs[16]. RORγt deletion causes a complete loss of ILC3s but not ILC1s or ILC2s. Runx3 is also required for the development of ILC1s and ILC3s, but not for ILC2s[6]. GATA3 is also involved in the development of ILC3s, and It continues to exert a critical role in mature ILC3s[49, 50]. These observations suggest that the development of different ILC subsets are controlled by TF networks[41]. Sall3 (Spalt-like transcription factor 3) belongs to the SAL family, which is implicated in embryonic development[36, 37, 51]. However, how Sall3 regulates the development of ILCs is still unknown. Here we define that Sall3 is a downstream target of IL-7 signaling, whose expression induced by IL-7Rα polyglutamylation drives CHILPs to differentiate ILC3s.

Glutamylation is highly conserved in all metazoans and protists, exerting critical roles in many physiological and pathological processes[52]. For example, TTLL7, the most abundantly expressed TTLLs in the mammalian nervous system, is conserved from acorn worm to primates, where it modulates neurite outgrowth and localization of dendritic MAPs[45]. ILC3s are enriched in Peyer's patches (PPs) and intestinal lamina propria[30]. Prior to the development of adaptive immunity, ILC3-induced IL-22 production has a critical function in priming innate immunity to eradicate *C. rodentium*[53, 54]. IL-22-deficient mice displayed exaggerated intestinal inflammation and impairment of the epithelial barrier and rapidly succumbed to bacterial infection. Given that ILC3s produce large amounts of IL-22, we thus used *C. rodentium* infection as a readout for determining the physiological function of ILC3s in the knockout mouse responses. The host protective effects of ILC3s are not restricted to bacterial infection in the intestine. ILC3s are also implicated in the

resistance to infections of *Candida albicans* and *Mycobacterium tuberculosis* in the lungs[55, 56]. Thereby ILC3s may be targeted to enhance or block immune responses for inflammatory pathology and immunotherapy. In this study, we show that the glutamylation and deglutamylation of IL-7Rα mediated by CCP2 and TTLL4/13 controls the development and effector function of ILC3s. Therefore, we strongly believe that it is necessary to develop specific inhibitors or agonists for these related polyglutamylases and CCPs. Manipulating polyglutamylation profiles by using these compounds, we may potentially target ILC3s for future clinical applications. In sum, IL-7Rα polyglutamylation has a critical function in the regulation of ILC3 development and their effector function. Our findings provide new mechanistic insights into how polyglutamylation modulates ILC3 development.

## Methods

**Antibodies and reagents.** Anti-CCP1 (LM-1A7), anti-CCP2 (S-13), anti-CCP3 (S-15), anti-CCP4 (T-17), anti-CCP5 (N-18), anti-CCP6 (N-14), anti-TTLL4 (S-14), anti-TTLL7 (E-12), anti-TTLL9 (C-20), anti-TTLL13 (D-16), anti-STAT4 (C-4), anti-P-STAT4 (E-2), anti-STAT6 (M-20), anti-P-STAT6 (sc-11762), anti-GST (6G9C6), and anti-Myc (9E10) were from Santa Cruz Biotechnology; Anti-STAT1 (14994), anti-P-STAT1 (9167), anti-STAT2 (72604), anti-STAT3 (9139), anti-P-STAT3 (9145), anti-H3K4me3 (9751), anti-STAT5 (9363), and anti-P-STAT5 (9351) were from Cell Signalling Technology (Danvers, USA). anti-IL-22 (135004) was from Biolegend. The antibodies against P-STAT2 (SAB4503836), Sall3 (SAB2102075), Flag-tag (M1), β-actin (SP124), and His-tag (6AT18) were from Sigma-Aldrich (St Louis, USA). GT335 antibody (AG-20B-0020) was from AdipoGen. Antibodies against CD3 (17A2), CD19 (1D3), B220 (RA3-6B2), IL-7Rα (A7R34), c-Kit (2B8), Sca-1 (D7), CD25 (PC61.5), CD11b (M1/70), CD11c (N418), Gr1 (RB6-8C5), F4/80 (BM8), Ter119 (TER-119), CD27 (LG.7F9), CD90 (HIS51), CD45.2 (104), RORγt (AFKJS-9), NKp46 (29A1.4), CD244 (C9.1), Flt3 (A2F10), $α_4β_7$ (DATK32), CD45.1 (A20), NK1.1 (PK136), IL-22 (IL22JOP), Thy1.2 (30-H12), and PLZF (Mags.21F7) were purchased from eBiosciences (San Diego, USA). Active caspase 3 antibody (550914) was purchased from BD Bioscience. All primary antibodies were used in a 1:2000 dilution for western blotting, in a 1:500 dilution for immunofluorescence staining and in a 1:100 dilution for flow cytometric staining. Paraformaldehyde (PFA, 158127), phenanthroline (Phen, 33510), CoCl2 (60818), and 4′,6-diamidino-2-phenylindole (DAPI; (D9542)) were from Sigma-Aldrich. IL-22 ELISA kit (BMS6022) was purchased from eBiosciences.

**Generation of knockout mice and Il7r^E446A mice.** *Ccp1* and *Ccp6* knockout mice were described previously[28]. *Ccp2^−/−*, *Ccp3^−/−*, *Ccp4^−/−*, *Ccp5^−/−*, *Ttll4^−/−*, and *Ttll13^−/−* mice were generated through CRISPR-Cas9 approaches as described[29]. Gt(ROSA)26Sor^tm1(CAG-xstpx-cas9,-EGFP)Fezh, *Rorc*(γt)^+/GFP and *Id2*^+/GFP mice were purchased from the Jackson Laboratory. *Stat3*^flox/flox was kindly provided by Dr Shizuo Akira (Osaka University, Japan). *Stat3*^flox/flox; *MxCre^+* mice were obtained by crossing *Stat3*^f/f mice with *MxCre^+* mice. To induce STAT3 deletion, 200 μg polyinosine-polycyticylic acid (poly(I:C)) was intraperitoneally injected to mice every other day for three times. Mouse experiments were performed according to the guidelines of the institutional animal care and use committees at the Institute of Biophysics, Chinese Academy of Sciences. For deletion of Sall3 in BM, B6;129-Gt(ROSA)26Sor^tm1(CAG-xstpx-cas9,-EGFP)Fezh knockin mice were crossed with Vav-Cre transgenic mice to generate *Rosa26-LSL-Cas9^+;Vav-Cre^+* mice. In all, $2 × 10^6$ BM cells were infected with lentiCRISPRv2 containing sgSall3 lentivirus. BM cells were then transplanted into lethally irradiated recipient mice (CD45.1^+). Donor-derived ILC3s were analyzed 8 weeks post transplantation. Sall3 deletion was confirmed by immunoblotting. For generation of *Il7r^E446A* mice, the genome locus of *Il7r* gene was knocked in with IL-7Rα-E446A mutation via a CRISPR-Cas9 approach. Mixture of Cas9 mRNA, single guide RNA (sgRNA), and IL-7Rα-E446A donor templates was microinjected into the cytoplasm of C57BL/6 fertilized eggs and transferred into the uterus of pseudopregnant ICR females. IL-7Rα-E446A mutations were identified by PCR screening and DNA sequencing. gRNA sequences are as follows: *Ccp2*: 5′-TAGAAATATTCTGGTTGATGTGG-3′; *Ccp3*: 5′-GGAGTATCAGCTAGGAAGATGGG-3′; *Ccp4*: 5′-AGCTCT-GAGCTGGTGCTCCCAGG-3′; *Ccp5*: 5′-GGTTCTACTTCAGTGTCCGGGG-3′; *Ttll4*: 5′-TTTGCCTCACGTTGGTGGCGG-3′; *Ttll13*: 5′-TTTCTTGGCTA-CAACCGATAAGG-3′; *Il7r*: 5′-TTCTTCTTGATTCAGTACTGAGG-3′; *Sall3*: 5′-CCAGCATCTCAAGTCGGACG-3′. Mice used in all experiments were 8-weeks old. And we performed three independent experiments of each mouse from at least three mice for each group. The background of mice was C57BL/6, and mice were grouped by the same age and gender. Animal use and protocols were approved by the Institutional Animal Care and Use Committees at Institute of Biophysics, Chinese Academy of Sciences.

**Histology analysis**. Mouse colons after *C. rodentium* infection were fixed in 4% PFA (Sigma-Aldrich) for 24 h, washed twice with phosphate-buffered saline (PBS) and stored using 75% ethanol before embedded in paraffin. Then colons in paraffin were sectioned and stained with hematoxylin and eosin (H&E) according to standard laboratory procedures.

**Intestinal lymphocyte separation**. Protocols for lymphocyte isolation from the intestine had been described[57]. With some modifications, intestines were dissected and cleaned, then PPs were removed. Intestines were cut longitudinally and washed with Dulbecco's Phosphate Buffered Saline (dPBS) five times. Then intestines were cut into pieces, and washed with solution I buffer (10 mM HEPES and 5 mM EDTA in Hank's Balanced Salt Solution (HBSS)) five times. For LPL isolation, the intestinal fragments were digested with solution II buffer containing DNaseI, 5% FBS, 0.2 mg/ml collagenase II and collagenase III there times at 37 °C. Then the tissues were sifted through 70-μm strainers.

**Flow cytometry**. BM cells were flushed out from femurs in PBS buffer and sifted through 70-μm cell strainers. For BM flow cytometric analysis, CLP (Lin⁻IL-7Rα⁺Sca-1$^{low}$c-Kit$^{low}$), $\alpha_4\beta_7^+$ precursor (Lin⁻IL-7Rα⁺c-Kit⁺$\alpha_4\beta_7^+$), CHILP (Lin⁻IL-7Rα⁺Flt3⁻CD25⁻Id2⁺$\alpha_4\beta_7^+$ or lin⁻IL-7Rα⁺Flt3⁻CD25⁻$\alpha_4\beta_7^+$ CD27⁺CD244⁺), ILCP (Lin⁻IL-7Rα⁺Id2⁺$\alpha_4\beta_7^+$PLZF⁺), ILC3 (Lin⁻RORγt⁺CD45⁺), and IL-22⁺ ILC3 (Lin⁻CD45⁺IL-22⁺) populations were analyzed or sorted with a FACSAria III instrument (BD Biosciences). Lineage cocktail antibody contains anti-B220, anti-CD3, anti-Ter119, anti-Gr1, anti-CD11b, anti-CD19, and anti-NK1.1. Data were analyzed using the FlowJo 7.6.1 software.

**Immunofluorescence assay**. Cells were isolated by fluorescence-activated cell sorting (FACS) and fixed with 4% PFA for 20 min at room temperature, then followed by 0.5% NP40 permeabilization and 10% donkey serum blocking. Cells were incubated with antibodies at 4 °C overnight, and then incubated with fluorescence-conjugated secondary antibodies. DAPI was used for nucleus staining. Cells were visualized by laser scanning confocal microscopy (Olympus FV1000, Olympus, Japan).

**Western blot**. Cells were lysed with RIPA buffer (150 mM NaCl, 0.5% sodium deoxycholate, 0.1% SDS, 1% NP40, 1 mM EDTA, 50 mM Tris (pH 8.0)), followed by separation with SDS-PAGE. Samples were then transferred to NC membrane and incubated with primary antibody in 5% bovine serum albumin (BSA). After washing with Tris Buffered Saline, with Tween-20 (TBST) three times, membranes were incubated with horseradish peroxidase (HRP)-conjugated secondary antibodies. At least $1 \times 10^5$ cells were used for a single sample for whole cell blots. See Supplementary Figs. 4 and 5 for uncropped blots.

**Quantitative real-time PCR**. Cell populations were isolated by FACS. And total RNAs were extracted with an RNA Miniprep Kit (Tiangen, Beijing, China) according to the manufacturer's protocol. Then complementary DNA (cDNA) was synthesized with the M-MLV reverse transcriptase (Promega, Madison, USA). mRNA transcripts were analyzed with ABI 7300 quantitative PCR (qPCR) system using specific primer pairs. Relative expressions were calculated and normalized to *Actb* expression. Primers for TTLLs and CCPs were described[28]. Other specific primers are as follows: *Il22* forward, 5′-TTGTGCGATCTCTGATGGCT-3′; *Il22* reverse, 5′-CCAGCATAAAGGTGCGGTTG-3′; *Twist2* forward: 5′-CAGAGC-GACGAGATGGACAA-3′; reverse: 5′-GAGAAGG CGTAGCTGAGACG-3′; *Sall3* forward: 5′-CCTGATTCTTCCTGGTGGAGT-3′; reverse: 5′-CTCTGGAAAA CGCCACAGAC-3′; *Hoxc8* forward: 5′-ACAGTAGCGAAG GACAAGGC-3′; reverse: 5′-CTTCAATCCGGCGCTTTCTG-3′; *Fosl1* forward: 5′-CCGGTCCA-CAGAGGTTCATC-3′; reverse: 5′-CTGGGCTGGATGTTCGGTAG-3′; *Pou4f1* forward: 5′-GCACACTGGGGAGCTGAG-3′; reverse: 5′-AAACGAACA AGGTGGGAGGG-3′; *Gapdh* forward: 5′-TGCACCACCAACTGCTTAG-3′; reverse: 5′-GGATGCAGGGATGATGTTC-3′.

**Chromatin immunoprecipitation assay**. CHILPs ($1 \times 10^5$) were isolated by FACS gating on Lin⁻IL-7Rα⁺Flt3⁻CD25⁻$\alpha_4\beta_7^+$ and cross-linked with 1% formaldehyde at 37 °C for 10 min. Then cells were washed twice with PBS, lysed and sonicated to get 300 to 500 bp DNA fragments. Lysates were incubated with 4 μg antibody overnight at 4 °C. Salmon sperm DNA/protein agarose beads were added for DNA immunoprecipitation. After washing, DNA was eluted from beads and purified. DNA fragments were analyzed using qPCR. Primers used for ChIP were as follows: *Sall3* forward: 5′-TAGTTCTCTTGCGCTCTTCCC-3′, reverse: 5′-CTTGGAA AGCTGTCTTTCGGT-3′.

**Recombinant protein preparation**. cDNAs were cloned from a BM cDNA library. CCP2 was subcloned into H-MBP-3c for MBP-tagged protein expression vectors. IL-7Rα was cloned into pGEX-6p-1 plasmid for GST-tagged protein expression. Plasmids were transformed into *E. coli* strain BL21 (DE3), followed by induction with 0.2 mM isopropyl-β-D-thiogalactoside (IPTG) at 16 °C for 24 h. Cells were collected and lysed by supersonic, followed by purification through Amylose or GST resins.

**EMSA assay**. EMSA experiments were conducted according to the manufacturer's protocol with a Light Shift Chemiluminescent RNA EMSA Kit (Thermo Scientific). Briefly, Flag-STAT5 was incubated with or without unlabeled probe for competitive reaction and anti-STAT5 antibody for super shift at room temperature for 20 min in a reaction buffer. Then, Biotin-labeled probe was added into the reaction system and incubated for 20 min at room temperature. Samples were carried out in 4% polyacrylamide gel in 0.5 × TBE buffer. After transferred on a nylon membrane (Amersham Biosciences), the labeled DNA was cross-linked by ultraviolet, probed with streptavidin-HRP conjugate and then incubated with the detection substrate. The probe sequence for *Sall3* was: 5′-CGGAGCCTAAAGCTGTTGCTTCGTG-GAACTTAGA CTAGCGGGAGAATTCAGTGTG-3′.

**DNase I accessibility assay**. DNaseI digestion assay has been described previously[28]. In brief, Nuclei were purified from CHILPs according to the manufacturer's protocol with the Nuclei isolating Kit (Sigma-Aldrich). Then nuclei were resuspended with DNase I digestion buffer and treated with indicated units of DNase I (Sigma, USA) at 37 °C for 5 min. In all, 2 × DNase I stop buffer (20 mM Tris Ph 8.0, 4 mM EDTA, 2 mM EGTA) was added to stop reactions. DNA was extracted and examined by qPCR.

**BM transplantation**. In all, $5 \times 10^4$ CD45.2⁺ LSK from *Ccp2*⁺/⁺ or *Ccp2*⁻/⁻ mice with $5 \times 10^6$ CD45.1⁺ helper cells were transplanted into lethally irradiated CD45.1⁺ recipients. Eight weeks after transplantation, percentages of ILC3s derived from donor cells were analyzed by FACS. For competitive transplantation, $1 \times 10^6$ CD45.2⁺ BM cells and $1 \times 10^6$ CD45.1⁺ BM cells were injected into lethally irradiated CD45.1⁺-recipient mice. Eight weeks after transplantation, ratios of CD45.2⁺ ILC3 to CD45.1⁺ ILC3 were analyzed. For IL-7Rα knockdown or over-expression, $1 \times 10^5$ CHILP cells were infected with retrovirus carrying shRNAs or overexpression sequences, followed by BM transplantation. Eight weeks after transplantation, chimeras were analyzed by FACS. Small interfering RNA sequences against IL-7Rα were cloned into LMP plasmid and IL-7Rα was cloned into pMY vector.

**Gene deletion in CHILPs by CRISPR/Cas9 technology**. Stat5 and Sall3 deletion in CHILPs were generated using Cas9 knockin mice according to the standard protocol provided by Zhang's lab[39]. Briefly, sgRNA was generated by online CRISPR Design Tool (http://tools.genome-engineering.org) and cloned into lentiCRISPRv2 for lentivirus production in 293 T cells. Then CHILPs were infected with lentivirus for *Stat5* or *Sall3* deletion. sgRNA sequences of *Stat5* were: *Stat5a*: 5′-AGGTAGTGCCGGACCTCGAT-3′; *Stat5b*: 5′-AAATAATGTCGCAC CTCGAT-3′.

**RNA interference**. Sequences for RNA interference were designed according to MSCV-LTRmiR30-PIG (LMP) system instructions. LMP vectors containing target sequences were constructed. The target sequence against IL-7Rα was: 5′-GCGTA TGTCACCATGTCTAGT-3′.

**In vitro ILC differentiation assay**. In vitro ILC differentiation assay was described previously[15]. In brief, CHILPs were sorted and cultured for 14 days on mitomycin C-treated OP9 feeder cells supplemented with IL-7 (25 ng/ml, Cat#: 217-17) and rSCF (25 ng/ml, Cat#: 250-03). Then cells were collected for flow cytometry.

**C. rodentium infection**. WT, *Ccp2*⁻/⁻, *Ccp6*⁻/⁻, *Ttll4*⁻/⁻, *Ttll13*⁻/⁻ and *Ttll4*⁻/⁻*Ttll13*⁻/⁻ mice were infected with $5 \times 10^9$ *C. rodentium* orally as described[6]. *C. rodentium* was a gift from Dr Baoxue Ge (Shanghai Institutes for Biological Sciences, Chinese Academy of Sciences). Mice were sacrificed by cervical dislocation to examine colon pathology and bacterial loads on day 8 post infection. Fecals, spleens, livers, blood, small intestines, and colons were collected from infected mice on day 8 after infection. Fecals, spleens, and livers were weighed and homogenized, and homogenates were plated on MacConkey agar plates for analysis of bacterial counts. Lamina propria lymphocytes (LPLs) were isolated from small intestines of infected mice, followed by analysis of ILC3s and IL-22.

**In vitro glutamylation assay**. Detailed protocol for in vitro glutamylation assay was described as previously described[28]. In brief, CCP2, TTLL4, and TTLL13 were transfected into 293 T cells for 48 h. Cells were harvested and lysed. Supernatants were incubated with GST-IL-7Rα at 37 °C for 2 h. GST-IL-7Rα was precipitated and tested for glutamylation with GT335 antibody.

**Statistical analysis**. An unpaired Student's *t*-test was used as statistical analysis in this study. Statistical calculation was performed by using Microsoft Excel or SPSS 13.

**Data availability**. All data generated or analyzed during this study are included in this published article and its Supplementary Information Files. Microarray data, are deposited in the Genebank as GSE97487.

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

## Acknowledgements

We thank Dr Dan Littman for providing pMIGR plasmid. We thank J.Li (Cnkingbio Company Ltd, Beijing, China) for technical support. We thank Jing Cheng, Jianhua Wang, Junfeng Hao, Yan Teng, Dongdong Fan, and Junying Jia for technical support. We thank Xiang Shi and Liangming Yao for technical help and assistance with animal procedures. This work was supported by the National Natural Science Foundation of China (31530093, 91640203, 31429001, 31670886, 31470864, 81572433, 81572433, 31601189, 81672956, and 81472413), the Strategic Priority Research Programs of the Chinese Academy of Sciences (XDB19030203, XDA12020219).

## Author contributions

B.L. designed and performed experiments, analyzed data, and wrote the paper; B.Y. performed experiments and analyzed data; G.H. performed some experiments; L.Y. crossed some mice; P.Z., J.W., and Y.D. analyzed data. X.Z., S.M., and T.Y. build up animal models and analyzed data; Z.F. initiated the study, organized, designed, and wrote the paper.

## Additional information

**Competing interests:** The authors declare no competing financial interests.

