## [Peer Review File · Nature Communications]

Reviewers' comments:

Reviewer #1 (Post-translational modification, inflammation)(Remarks to the Author):

Liu et al., described "glutamylation of IL-7Ra controls the specification of group 3 ILCs by activation of Sall3. Authors used several different strains of knockout mice such as carboxypeptidase (CCP) family responsible for deglutamylation of cytosolic proteins and tubulin glutamyl ligases (TTLL) family responsible for glutamylation. Authors found that CCP2 and TTLL4 or TTLL13 are specifically associated with the polyglutamylation of the residue E446 in intracellular domain of IL-7Ra. The knock-in experiment of murine IL-7Ra/E446A mutant impaired Sall3 expression and ILC3 lineage development. Interestingly, both human and mouse residue E446 of IL-7Ra in intracellular domain are conserved. Previous report by the same group in 2016 suggested the deficiency in the CCP5 or CCP6 led to susceptibility to viral infection due to polyglutamylation of cGAS by the enzyme TTLL6 impeded its DNA binding while CCP6 deglutamylation of cGAS led to the activation of cGAS. Therefore, glutamylation and deglutamylation of cGAS tightly modulate immune responses in viral infection.

In this study, authors were able to show each step of IL-7Ra glutamylation and deglutamylation with various gene deficient mice with proper control mice from same family gene besides signaling molecule such as STAT5 deficient mouse, which is an essential in downstream of IL-7 signaling. The interpreted data are presented with appropriate methods. It is well established that IL-7 is very important in the development of lymphoid lineages and this fact has been shown with IL-7 and IL-7Ra deficient mouse. Most of these studies have been completed before the characterization of ILC lineages. It is possible that there is a specific role of IL-7 in ILC3 lineage development and it is worth to investigate specific role of IL-7/IL-7Ra in ILC3 lineage development. This study may explain the importance of IL-7/IL-7Ra/STAT5/Sall3 axis in the development of ILC3 lineage and host defense against gastrointestinal infection. The Spalt like transcription factor 3 (Sall3) is probably a new player in ILC3 development.

Prior to publishing these data, a major concern should be addressed. It is necessary to resolve whether the polyglutamylation of IL-7Ra E446 residue is happened by direct stimulation with the specific ligand such as IL-7 and TSLP. If IL-7Ra E446 residue polyglutamylation is not occurred by IL-7 or TSLP stimulation, the observed polyglutamylation on IL-7Ra is independent of ligand stimulation. If this is a case, the title of manuscript must be changed. However author used the protocol of in vitro ILC differentiation assay to culture CHILPs for 14 days on mitomycin C-treated OP9 feeder cells supplemented

with IL-7 (25 ng/ml) and rSCF (25 ng/ml). Author need to write the sources of these cytokines or vendor if they purchase them. This must be addressed in main text at the beginning of manuscript because it is most important factor in ILC3 differentiation. None will pay attention to read the material and methods part and figure out how author cultured the ILC3 type cells. Suggest using TSLP, another ligand for IL-7Ra to test whether TSLP induces ILC3 differentiation similar to IL-7 activity.

The mechanisms of ILC1 and ILC3 balance during colitis remain unknown. However few previous studies suggested that these regulations are controlled by IL-7, IL-15, and IL-2. ILC development, these cytokines play major roles and further investigation on IL-7, IL-15, and IL-2 including their receptors will help understand ILC lineage development. Cite this paper (Transcriptional Programs Define Molecular Characteristics of Innate Lymphoid Cell Classes and Subsets; Nat Immunol. 2015 March; 16(3): 306–317. doi:10.1038/ni.3094).

Authors summarize the data in a schematic figure that will help understand the large number of different KO mice and of results used in this manuscript. IL-7/TSLP → IL-7Ra (put the involvement of glutamylation enzymes; Ttll4 or Ttll13 KO/CCP2 KO cascade while IL-7Ra synthesis is intracellular) → STAT5 activation → Sall3 → ILC3 development.

1. When abbreviating a term, use the full name the first time you use it in the beginning of each section such as abstract and introduction.
2. The expression of "largely unknown" in the sentence is inappropriate. Please change it.
3. In line 8, page 5, the term "CCP mutant mouse strain" is must be changed to "CCP deficient mouse or KO mouse". Mutant means a point mutation or a partial deletion of gene unlike a complete deletion. Find this matter throughout main text and figure legends.
4. In line 2 from bottom, page 6, check spelling "CCP2-deficient BM lysates" throughout manuscript!
5. In line 9 from top and 1 from bottom, page 8, use Italic letter "in vitro". Check throughout manuscript.
6. In line 2, page 9, suggest "ILC3 specification" is replaced with ILC3 development.
7. In page 9, author must address the STAT5 activation in the absence of IL-7 and TSLP, ligand of IL-17Ra in the Figure 5A-C.
8. In Figure 5E, the expression of Sall3 is dependent on STAT5 phosphorylation

Author must address the STAT5 activation in the absence or presence of IL-7 and TSLP, ligand for IL-17Ra. What you found IL-7Ra hyperglutamylation that is coincident without IL-17Ra ligand functional connection! Or authors perform quick experiments that treat wild type and IL-7Ra E446 mutant cells to show hyperglutamylation only in IL-7Ra WT. If this is not case, you should change the title, removing "Glutamylation of IL-7Ra controls" in title, which is suggested before.

9. In line 9-12, page 11, change below sentence. It is repeated!

In this study, we show that CCP2 is highly expressed in CHILPs, whose deficiency causes increased numbers of ILC3s. We identify IL-7Ra is a new substrate of CCP2, whose polyglutamylation is catalyzed by TTLL4 and TTLL13 in CHILPs.

Reviewer #2 (Innate lymphoid cells, transcription regulation)(Remarks to the Author):

This manuscript is very interesting. It aims to delineate whether carboxypeptidases play a role in the development or maintenance of ILCs. The authors have analysed different CCP to identify that CCP2 is highly expressed in the ILC progenitor.

The focus of the work is on ILC3 but it is unclear if this is solely limited to ILC3.

1. What is the effect of CCP2 loss in other ILC populations? This question refers to Figure 1 that only shows ILC3 and Figure 2A that also only shows ILC3. There would be both interest in whether CCP loss affects other ILCs and whether more ILC3 influences the balance between other ILCs esp ILC2.
2. The report focuses on reporting enhanced ILC3 populations and reporting these as a percentage – is this similar/different for the cell numbers? None are currently shown.
3. Figure 2B. The staining for the CLP is quite poor. Again, everything is reported in percentage but it is not clear that the compartments are actually equal to the WT.
4. Given this pathway affects IL7R, is there not a phenotype in T cells? And dendritic cells which express high levels of CCP2? Such defects are likely to impact considerably on the outcome for the citrobacter experiments.
5. Is it clear that IL-22 is not regulated by loss of CCP2? ie. Is the change in IL-22 due to change in cell number or actual increase in IL-22 per cell.
6. If CCP2 affects CHILPS then it is unclear why it only subsequently affects ILC3s. Could the authors clarify?
7. The experiments on Sall3 are interesting (Fig. 5N). It would be helpful to see mixed chimeras from these mice; what other cells are dysregulated in the absence of Sall3?
8. Figure 2B is very poor quality staining.
9. The link to IL-7R seems to be a jump based solely on Figure 2I?? While this tracks to a nice lead, it remains unclear how this candidate was arrived at esp given that even amongst the limited phenotyping done, equal changes occurred in other markers??
10. Figure 2A is unreadable and needs to be larger.
11. Could the authors provide a fuller analyses/table for the microarray data shown in Figure 5D. Currently the heatmap does not provide any substantive information although a small portion is expanded in Figure 5E. It would be preferable to provide the full list for the reader.
12. Could a scale be provided for Figure 6E.
13. Could the authors clarify what the error bars represent – SD or SEM?
14. The assumption is that changes in the knockout mouse responses to Citrobacter are attributable to ILC3 – there is no evidence that the outcome is solely directed by ILC3 rather than also other immune cells affected.

Reviewers' comments:

Reviewer #2 (Remarks to the Author):

This work is substantially improved though some outstanding issues remain.

The authors have now included a modified sentence: "Since we used the same cell numbers of WT and Ccp2^{-/-} ILC3s for these above assays, we thus conclude that the change in IL-22 is due to both cell number and actual increase in IL-22 per cell." (Line 135) This rationale does not make sense and requires clarification as the outcome would not reflect an increase in cell number based on these data. This entire paragraph contains the appropriate message but is linked to incorrect pieces of the data. It requires revision.

The authors have added that "increased numbers of ILC3s, but reduced numbers of ILC1s and ILC2s (data not shown)"
- this statement should be moved to the results rather than appear in the discussion. It is appreciated that further work is being conducted on this aspect but it is important that the paper be an accurate reflection of the deficient mouse.

It was indicated that mixed chimeric mice should be analysed. This does not appear to have been performed and is quite different from analyses of complete knockout mice. It is partly overcome by examining Figure 2j which addresses whether the effects are intrinsic (or not) but does not determine whether the effect is indeed driven by IL7R which remains correlative rather than proven by these experiments.

The authors still argue that IL-22 largely produced by ILC3 are responsible for the phenotype observed in their knockout mice. However, previous work in which IL-22 is selectively deleted using NCRiCre indicate that such loss in ILC3 may have little impact and that the effect is substantially attributable to other cell types. It is thus unclear how the complete knockout mouse can test the role in this setting.

The English needs considerable improvement.

REVIEWERS' COMMENTS:

Reviewer #2 (Remarks to the Author):

The manuscript is improved by the corrections and addition of data.

I have only one remaining concern which is around the wording for the interpretation of that CCP2 is responsible totally for the phenotype - it could be suggested that the experiments infer this as a likely proposition but given the effects on other cell types from a knockout, the involvement of other immune cells cannot be excluded (a similar situation occurred in the Guo X Immunity 2014 paper where RorgtCre affects both T cells and ILC3). This is relatively easily remedied but a slight alteration of the wording to indicate the potential involvement of other cells cannot be excluded. Nevertheless, this is an exciting piece of work.

Point-by-point response to reviewers' comments

Reviewer #1:

Liu et al., described "glutamylation of IL-7R α controls the specification of group 3 ILCs by activation of Sall3. Authors used several different strains of knockout mice such as carboxypeptidase (CCP) family responsible for deglutamylation of cytosolic proteins and tubulin glutamyl ligases (TTLL) family responsible for glutamylation. Authors found that CCP2 and TTLL4 or TTLL13 are specifically associated with the polyglutamylation of the residue E446 in intracellular domain of IL-7R α . The knock-in experiment of murine IL-7R α /E446A mutant impaired Sall3 expression and ILC3 lineage development. Interestingly, both human and mouse residue E446 of IL-7R α in intracellular domain are conserved. Previous report by the same group in 2016 suggested the deficiency in the CCP5 or CCP6 led to susceptibility to viral infection due to polyglutamylation of cGAS by the enzyme TTLL6 impeded its DNA binding while CCP6 deglutamylation of cGAS led to the activation of cGAS. Therefore, glutamylation and deglutamylation of cGAS tightly modulate immune responses in viral infection.

In this study, authors were able to show each step of IL-7R α glutamylation and deglutamylation with various gene deficient mice with proper control mice from same family gene besides signaling molecule such as STAT5 deficient mouse, which is an essential in downstream of IL-7 signaling. The interpreted data are presented with appropriate methods. It is well established that IL-7 is very important in the development of lymphoid lineages and this fact has been shown with IL-7 and IL-7R α deficient mouse. Most of these studies have been completed before the characterization of ILC lineages. It is possible that there is a specific role of IL-7 in ILC3 lineage development and it is worth to investigate specific role of IL-7/IL-7R α in ILC3 lineage development. This study may explain the importance of IL-7/IL-7R α /STAT5/Sall3 axis in the development of ILC3 lineage and host defense against gastrointestinal infection. The spalt like transcription factor 3 (Sall3) is probably a new player in ILC3 development.

Prior to publishing these data, a major concern should to be addressed. It is necessary to resolve whether the polyglutamylation of IL-7R α E446 residue is happened by direct stimulation with the specific ligand such as IL-7 and TLSP. If IL-7R α E446 residue polyglutamylation is not occurred by IL-7 or TLSP stimulation, the observed polyglutamylation on IL-7R α is independent of ligand stimulation. If this is a case, the title of manuscript must be changed. However author used the protocol of in vitro ILC differentiation assay to culture CHILPs for 14 days on mitomycin C-treated OP9 feeder cells supplemented with IL-7 (25 ng/ml) and rSCF (25 ng/ml). Author need to write the sources of these cytokines or vendor if they purchase them. This must be addressed in main text at the beginning of manuscript because it is most important factor in ILC3 differentiation. None will pay attention to read the material and methods part and figure out how author cultured the ILC3 type cells. Suggest using TSLP, another ligand for IL-7R α to test whether TSLP induces ILC3 differentiation similar to IL-7 activity.

The mechanisms of ILC1 and ILC3 balance during colitis remain unknown. However few previous studies suggested that these regulations are controlled by IL-7, IL-15, and IL-2. ILC development, these cytokines play major roles and further investigation on IL-7, IL-15, and IL-2 including their receptors will help understand ILC lineage development. Cite

this paper (Transcriptional Programs Define Molecular Characteristics of Innate Lymphoid Cell Classes and Subsets; Nat Immunol. 2015 March; 16(3): 306–317. doi:10.1038/ni.3094).

Answer: This is a very good point. As shown in the new Fig. 5p, IL-7R α polyglutamylation indeed occurred in WT CHILPs with IL-7 stimulation, and STAT5 phosphorylation appeared as well. However, in the absence of IL-7, IL-7R α did not undergo polyglutamylation and no STAT5 phosphorylation appeared in WT CHILPs. By contrast, in the presence of IL-7, CHILPs from IL-7R α E446A mutation knockin (Il7r^{E446A}) mice did not undergo polyglutamylation and no STAT5 phosphorylation appeared either. Similar results were obtained with TSLP treatment (data not shown). For the vendor of these cytokines we used in our culture system, murine IL-7 (Cat#: 217-17) and SCF (Cat#: 250-03) were purchased from Peprotech, USA. We also used TSLP for ILC3 differentiation in our culture system, similar results were obtained as those of IL-7 treatment. We addressed this issue in the revised text. We cited this paper and stated their findings in our revised discussions.

Authors summarize the data in a schematic figure that will help understand the large number of different KO mice and of results used in this manuscript. IL-7/TSLP → IL-7Ra (put the involvement of glutamylation enzymes; Tll4 or Tll13 KO/CCP2 KO cascade while IL-7Ra synthesis in intracellular) → STAT5 activation → Sall3 → ILC3 development.

Answer: We made a schematic figure as shown in the new Fig. 6k.

Minor comments:

1. When abbreviating a term, use the full name the first time you use it in the beginning of each section such as abstract and introduction.

Answer: We revised our manuscript as suggested.

2. The expression of “largely unknown” in the sentence is inappropriate. Please change it.

Answer: We changed it.

3. In line 8, page 5, the term “CCP mutant mouse strain” is must be changed to “CCP deficient mouse or KO mouse”. Mutant means a point mutation or a partial deletion of gene unlike a complete deletion. Find this matter throughout main text and figure legends.

Answer: We changed it throughout our revised manuscript.

4. In line 2 from bottom, page 6, check spelling “CCP2-deficient BM lysates” throughout manuscript!

Answer: We corrected it thoroughly.

5. In line 9 from top and 1 from bottom, page 8, use Italic letter “in vitro”. Check throughout manuscript.

Answer: We revised it in our revised text.

6. In line 2, page 9, suggest "ILC3 specification" is replaced with ILC3 development.

Answer: We changed it.

7. In page 9, author must address the STAT5 activation in the absence of IL-7 and TSLP, ligand of IL-17Ra in the Figure 5A-C.

Answer: We addressed this issue as the answer of the first major concern (new Fig. 5p). We stated these results in our revised text.

8. In Figure 5E, the expression of Sall3 is dependent on STAT5 phosphorylation. Author must address the STAT5 activation in the absence or presence of IL-7 and TSLP, ligand for IL-7Ra. What you found IL-7Ra hyperglutamylation that is coincident without IL-7Ra ligand functional connection! Or authors perform quick experiments that treat wild type and IL-7Ra E446 mutant cells to show hyperglutamylation only in IL-7Ra WT. If this is not case, you should change the title, removing "Glutamylation of IL-7Ra controls" in title, which is suggested before.

Answer: This is a good suggestion. We addressed this issue as the answer of the first major concern (new Fig. 5p).

9. In line 9-12, page 11, change below sentence. It is repeated!

In this study, we show that CCP2 is highly expressed in CHILPs, whose deficiency causes increased numbers of ILC3s. We identify IL-7R α is a new substrate of CCP2, whose polyglutamylation is catalyzed by TLL4 and TLL13 in CHILPs.

Answer: We revised this sentence.

Reviewer #2:

This manuscript is very interesting. It aims to delineate whether carboxypeptidases play a role in the development or maintenance of ILCs. The authors have analysed different CCP to identify that CCP2 is highly expressed in the ILC progenitor.

The focus of the work is on ILC3 but it is unclear if this is solely limited to ILC3.

1. What is the effect of CCP2 loss in other ILC populations? This question refers to Figure 1 that only shows ILC3 and Figure 2A that also only shows ILC3. There would be both interest in whether CCP loss affects other ILCs and whether more ILC3 influences the balance between other ILCs esp ILC2.

Answer: This is a very good point. Actually, we also analyzed ILC1s and ILC2s in the small intestine of *Ccp2*^{-/-} mice. We found that the numbers of ILC1s and ILC2s were decreased in *Ccp2*-deficient mice (data not shown). We are still investigating the molecular mechanisms by which how CCP2 regulates the development of other ILCs. We addressed this issue in the discussion section. However, how CCP2-mediated IL-7 signaling regulates the switch balance of ILC development is beyond the scope of this paper.

2. The report focuses on reporting enhanced ILC3 populations and reporting these as a percentage – is this similar/different for the cell numbers? None are currently shown.

Answer: This is the case. The number of ILC3s in *Ccp2*-deficient mice was also

increased. We added these calculation data in respective figures (Fig. S1c).

3. Figure 2B. The staining for the CLP is quite poor. Again, everything is reported in percentage but it is not clear that the compartments are actually equal to the WT.

Answer: We replaced this figure with better images. We showed cell numbers as well.

4. Given this pathway affects IL7R, is there not a phenotype in T cells? And dendritic cells which express high levels of CCP2? Such defects are likely to impact considerably on the outcome for the citrobacter experiments.

Answer: This is a good point. IL-7/IL-7R signaling is also required for the regulation of T cells and DCs as well. However, the regulations of IL-7/IL-7R signaling cascades must be distinct in various kinds of cell types. We checked CCP2 expression levels in T cells and DCs. We observed that CCP2 was almost undetectable in these two cell types (data not shown). In addition, we found that CCP2 deficient mice did not affect the numbers of T cells and DCs in bone marrow and peripheral blood (data not shown). These results suggest that IL-7R α polyglutamylation does not influence the development of T cells and DCs. However, we cannot exclude the possibility that such defects of the knockout mouse responses to *C. rodentium* infection might be synergistically modulated by adaptive immune responses mediated by T cells or DCs.

5. Is it clear that IL-22 is not regulated by loss of CCP2? ie. Is the change in IL-22 due to change in cell number or actual increase in IL-22 per cell.

Answer: For the Fig. 1i, we analyzed *I22* mRNA levels in equal numbers of WT and *Ccp2*^{-/-} ILC3s. As for the Fig. 1j, we used the same cell numbers of WT and *Ccp2*^{-/-} ILC3s with IL-23 stimulation. Based on these results, CCP2 deficiency increases IL-22 production per cell as well. Taken together, we conclude that the change in IL-22 is due to both cell number and actual increase in IL-22 per cell. However, it still needs to be further investigated how CCP2 regulates IL-22 production in ILC3s. This is beyond the scope of this paper.

6. If CCP2 affects CHILPs then it is unclear why it only subsequently affects ILC3s. Could the authors clarify?

Answer: This issue was addressed as the answer of Question#1. We addressed this issue in the discussion section.

7. The experiments on Sall3 are interesting (Fig. 5N). It would be helpful to see mixed chimeras from these mice; what other cells are dysregulated in the absence of Sall3?

Answer: We analyzed all ILCs in Sall3-deficient mice. We observed that Sall3-deficient mice exhibited reduced ILC3s. By contrast, Sall3-deficient mice did not significantly affect the numbers of ILC1s and ILC2s (data not shown). We are still investigating about how Sall3 regulates other immune cells.

8. Figure 2B is very poor quality staining.

Answer: We replaced this figure with better images.

9. The link to IL-7R seems to be a jump based solely on Figure 2I?? While this tracks to a nice lead, it remains unclear how this candidate was arrived at esp given that even amongst the limited phenotyping done, equal changes occurred in other markers??

Answer: For Figure 2i, we tested the changes of ILCPs post competitive transplantation. The results of this figure with the Figures 2h and 2j represented an increased number of ILC3s caused by CCP2 deficiency in an intrinsic manner. For ILCP analysis, there were no other markers available up to date.

10. Figure 2A is unreadable and needs to be larger.

Answer: We changed it.

11. Could the authors provide a fuller analyses/table for the microarray data shown in Figure 5D. Currently the heatmap does not provide any substantive information although a small portion is expanded in Figure 5E. It would be preferable to provide the full list for the reader.

Answer: We provided the full list of differential transcription factors in *Ccp2*^{-/-} CHILPs.

12. Could a scale be provided for Figure 6E.

Answer: We added a scale bar for this figure.

13. Could the authors clarify what the error bars represent – SD or SEM?

Answer: The error bars represent SD. We added these messages in the according legends.

14. The assumption is that changes in the knockout mouse responses to *Citrobacter* are attributable to ILC3 – there is no evidence that the outcome is solely directed by ILC3 rather than also other immune cells affected.

Answer: This is a very good point. As we know, many immune cells and a variety of cytokines participate in the clearance of *C. rodentium*. Accumulating evidence shows that IL-22 plays a critical role in the early host defense against *C. rodentium*. Given that ILC3s produce large amounts of IL-22, we thus used *C. rodentium* infection as a readout for determining the physiological function of ILC3s in the knockout mouse responses. Surely, in our current study, we cannot exclude other immune cells are implicated in the knockout mouse responses to *C. rodentium* infection. We addressed this issue in the discussion section of our revised manuscript.

Point-by-point response to the reviewer #2

Reviewer #2:

This work is substantially improved though some outstanding issues remain. The authors have now included a modified sentence: "Since we used the same cell numbers of WT and *Ccp2*^{-/-} ILC3s for these above assays, we thus conclude that the change in IL-22 is due to both cell number and actual increase in IL-22 per cell."(Line 135) This rationale does not make sense and requires clarification as the outcome would not reflect an increase in cell number based on these data. This entire paragraph contains the appropriate message but is linked to incorrect pieces of the data. It requires revision.

Answer: We deleted this wrong conclusion. We replaced the old Fig. 1i with new data. In the new Fig. 1i, we isolated ILC3s from WT, *Ccp2*^{-/-} or *Ccp6*^{-/-} intestines and incubated with IL-23, followed by examination of IL-22 secretion. We carefully revised this paragraph.

The authors have added that "increased numbers of ILC3s, but reduced numbers of ILC1s and ILC2s (data not shown)". - this statement should be moved to the results rather than appear in the discussion. It is appreciated that further work is being conducted on this aspect but it is important that the paper be an accurate reflection of the deficient mouse.

Answer: This is a good suggestion. We added these data in the new Supplementary Fig. 1d, e, and stated these results in the main text.

It was indicated that mixed chimeric mice should be analysed. This does not appear to have been performed and is quite different from analyses of complete knockout mice. It is partly overcome by examining Figure 2j which addresses whether the effects are intrinsic (or not) but does not determine whether the effect is indeed driven by IL7R which remains correlative rather than proven by these experiments.

Answer: This is a very good point. We already conducted BM transplantation to verify the role of IL-7R in CCP2-mediated ILC3 development. We engrafted 5×10^4 CD45.2⁺ LSKs from WT or *Il7r*^{E446A} mice with 5×10^6 CD45.1⁺ helper cells into lethally irradiated CD45.1⁺ recipients for eight weeks, followed by analysis of mixed chimeras. As shown in the new Supplementary Fig. 2f, *Il7r*^{E446A} BM transferred recipients displayed a reduced number of ILC3s compared to those of WT BM engraftment. We stated this result in the revised text.

The authors still argue that IL-22 largely produced by ILC3 are responsible for the phenotype observed in their knockout mice. However, previous work in which IL-22 is selectively deleted using NCRiCre indicate that such loss in ILC3 may have little impact and that the effect is substantially attributable to other cell types. It is thus unclear how the complete knockout mouse can test the role in this setting.

Answer: In addition to ILC3s, other immune cells can also produce IL-22. ILC3 cells are heterogeneous and can be further subclassified into additional subsets according to their co-receptor CD4 and the NKp46 (encoded by *Ncr1* gene), including CD4⁻NKp46⁺ ILC3s, CD4⁻NKp46⁻ ILC3s and CD4⁺NKp46⁻ ILC3s (Ebihara T, et al. Nat Immunol, 2015; Lee J, et al. Nat Immunol, 2011). Using NCRiCre only deletes the subset of CD4⁻NKp46⁺

ILC3s, but has no impact on other subsets of NKp46⁻ ILC3s. It has been reported that IL-22-producing ROR γ ⁺ ILCs are essential for resistance to *C. rodentium* infection (Guo X, et al. Immunity, 2014). In our study, CCP2 knockout causes increase of total ILC3s and IL-22 production, leading to resistance to *C. rodentium* infection. Surely, we cannot exclude whether other cell types are regulated by CCP2 that also involved in the resistance to *C. rodentium* infection. This is beyond the scope of this study.

The English needs considerable improvement.

Answer: We carefully revised our whole manuscript.

Point-by-point response to the reviewer #2

Reviewer #2:

The manuscript is improved by the corrections and addition of data.

I have only one remaining concern which is around the wording for the interpretation of that CCP2 is responsible totally for the phenotype - it could be suggested that the experiments infer this as a likely proposition but given the effects on other cell types from a knockout, the involvement of other immune cells cannot be excluded (a similar situation occurred in the Guo X Immunity 2014 paper where RorgtCre affects both T cells and ILC3). This is relatively easily remedied but a slight alteration of the wording to indicate the potential involvement of other cells cannot be excluded. Nevertheless, this is an exciting piece of work.

Answer: I agree to this point. I addressed this issue in the discussion section of our revised manuscript.